# Antimicrobial Activity of *Bacillus amyloliquefaciens* BS4 against Gram-Negative Pathogenic Bacteria

**DOI:** 10.3390/antibiotics13040304

**Published:** 2024-03-28

**Authors:** Ana Paula Palacios-Rodriguez, Abraham Espinoza-Culupú, Yerson Durán, Tito Sánchez-Rojas

**Affiliations:** 1Laboratory of Environmental Microbiology and Biotechnology, Faculty of Biological Sciences, Universidad Nacional Mayor de San Marcos, Lima 15081, Peru; ana.palacios@unmsm.edu.pe (A.P.P.-R.); yerson.duran@unmsm.edu.pe (Y.D.); 2Laboratory of Molecular Microbiology and Biotechnology, Faculty of Biological Sciences, Universidad Nacional Mayor de San Marcos, Lima 15081, Peru

**Keywords:** antimicrobial activity, *Bacillus amyloliquefaciens* BS4, culture conditions, broad-spectrum metabolites

## Abstract

Worldwide, bacterial resistance is one of the most severe public health problems. Currently, the failure of antibiotics to counteract superbugs highlights the need to search for new molecules with antimicrobial potential to combat them. The objective of this research was to evaluate the antimicrobial activity of *Bacillus amyloliquefaciens* BS4 against Gram-negative bacteria. Thirty yeasts and thirty-two *Bacillus* isolates were tested following the agar well-diffusion method. Four *Bacillus* sp. strains (BS3, BS4, BS17, and BS21) showed antagonistic activity against *E. coli* ATCC 25922 using bacterial culture (BC) and the cell-free supernatant (CFS), where the BS4 strain stood out, showing inhibitory values of 20.50 ± 0.70 mm and 19.67 ± 0.58 mm for BC and CFS, respectively. The *Bacillus* sp. BS4 strain can produce antioxidant, non-hemolytic, and antimicrobial metabolites that exhibit activity against several microorganisms such as *Salmonella enterica*, *Klebsiella pneumoniae*, *Shigella flexneri*, *Enterobacter aerogenes*, *Proteus vulgaris*, *Yersinia enterocolitica*, *Serratia marcescens*, *Aeromonas* sp., *Pseudomonas aeruginosa*, *Candida albicans*, and *Candida tropicalis*. According to the characterization of the supernatant, the metabolites could be proteinaceous. The production of these metabolites is influenced by carbon and nitrogen sources. The most suitable medium to produce antimicrobial metabolites was TSB broth. The one-factor-at-a-time method was used to standardize parameters such as pH, agitation, temperature, carbon source, nitrogen source, and salts, resulting in the best conditions of pH 7, 150 rpm, 28 °C, starch (2.5 g/L), tryptone (20 g/L), and magnesium sulfate (0.2 g/L), respectively. Moreover, the co-culture was an excellent strategy to improve antimicrobial activity, achieving maximum antimicrobial activity with an inhibition zone of 21.85 ± 1.03 mm. These findings position the *Bacillus amyloliquefaciens* BS4 strain as a promising candidate for producing bioactive molecules with potential applications in human health.

## 1. Introduction

The discovery of antibiotics as a treatment for numerous diseases marked a crucial milestone in medicine and stands as one of the most notable advancements of the 20th century [1]. However, the overuse or misuse of antibiotics in both animals and humans, environmental pollution, and ineffective infection control measures are major contributors to bacterial resistance, posing significant global public health challenges. This issue is recognized as a critical concern within the One Health framework [2,3,4], further exacerbated by the limited development of new antimicrobial agents [1].

Despite concerted efforts to combat bacterial resistance, its trajectory remains unabated [4]. As a result, infections stemming from resistant bacteria elicit more pronounced symptoms, necessitate prolonged hospitalizations, and exacerbate morbidity and mortality rates among patients in both developed and developing nations [5]. This increase in healthcare burden leads to a substantial increase in associated costs [6,7]. One of the most alarming scenarios involves the collapse of healthcare systems due to the unavailability of antibiotics, resulting in harm to patients undergoing surgical procedures, organ transplants, and chemotherapy regimens [1].

The ineffectiveness of antibiotics against resistant bacteria underscores the necessity to discover new antimicrobial molecules to address the global challenge of resistance [8]. Among the promising candidates under investigation are antimicrobial metabolites synthesized by bacteria of the genus *Bacillus* [9] and yeasts. The *Bacillus* genus is distinguished by its prolific production of diverse substances, including ribosomal and non-ribosomal peptides, volatile compounds, polyketides, and hybrids of polyketides and non-ribosomal peptides [5]. Conversely, yeasts exhibit an antagonistic effect against pathogenic microorganisms through various mechanisms, such as nutrient competition, organic or volatile acid production, high ethanol concentrations, secretion of antibacterial compounds, and toxin release [10,11].

Conventional approaches to microbial research typically involve the extraction and isolation of bioactive compounds from fermentation broth. However, these methods are often inefficient, leading to the rediscovery of known bioactive molecules [6]. In response, scientists have increasingly turned to genomic mining as a powerful tool for discovering new metabolites. By studying biosynthetic gene clusters that encode metabolic pathways, genomic mining provides valuable and objective insights into natural product biosynthesis [7]. Nevertheless, it is widely recognized that a significant proportion of biosynthetic gene clusters are dormant or silent under standard laboratory fermentation conditions [12]. Consequently, only a fraction of the potential chemical structures encoded by these gene clusters are actually produced.

These dormant genetic loci are commonly termed “cryptic” or “orphan”. To unlock the potential of these untapped reservoirs of potentially bioactive compounds, the biosynthesis of these presumed metabolites must be induced. Moreover, under these standard conditions, microbial secondary metabolites may remain undetected due to low production rates, substantial metabolic background, or unfavorable culture conditions [6]. Hence, the optimization of culture conditions has been recognized as the simplest and most effective strategy to enhance the production of microbial secondary metabolites, a concept known as OSMAC (“one strain, many compounds”), terminology introduced by Zeeck et al. [8].

Microbial fermentation stands as a fundamental process for enhancing the production of a broad spectrum of metabolites essential to the pharmaceutical industry. Maximum production yields are attained through the optimization of culture parameters, including nutrient supply, agitation, pH, temperature, inoculum concentration, and incubation period [9]. By strategically adjusting these easily accessible culture parameters, researchers have successfully isolated numerous previously unknown natural products from various fungi and actinomycetes [6]. The objective of this research was to evaluate the antimicrobial activity of *Bacillus amyloliquefaciens* BS4 against Gram-negative bacteria and investigate the effect of culture conditions on enhancing the production of antimicrobial metabolites.

## 2. Results

### 2.1. Screening of Antagonistic Yeast

A screening test of 30 native yeast strains was performed to determine their antagonistic activity against *Escherichia coli* ATCC 25922. None of the 30 yeast strains exhibited such activity (Appendix A).

### 2.2. Screening of Antagonistic Bacteria

A screening test of 32 *Bacillus* isolates for their antagonistic activity against *E. coli* ATCC 25922 was performed. The results revealed that four strains exhibited antagonistic activity, as determined using the agar well-diffusion method. Further details are provided in Appendix A.

According to the results, the antagonistic activity against *E. coli* ATCC 25922 using bacterial culture (CB) and cell-free supernatant (CFS) of the *Bacillus* strains remained consistent over the 72 h period. The BS4 strain exhibited the highest activity in both cases, as shown in Table 1. Consequently, this strain was selected for further experimentation. 

The *Bacillus* sp. BS4 strain displayed specific colony characteristics, appearing translucent, mucous, and smooth after 24 h of incubation. By 48 h, the colonies became rough and opaque while retaining their mucous appearance (Figure 1a). Microscopic examination revealed the presence of Gram-positive bacilli (Figure 1b). 

### 2.3. Effect of Different Culture Conditions on Enhancing Antimicrobial Activity of the Bacillus sp. BS4 Strain

#### 2.3.1. Selection of Fermentation Medium 

The effect of the three culture broths (TSB, LB, and YPG) on bacterial growth and antimicrobial activity at various time intervals is depicted in Table 2. In terms of bacterial growth, similar biomass values were observed between the three culture broths (TSB, LB, and YPG).

The cell-free supernatant from the TSB broth exhibited the highest antimicrobial activity against *E. coli* ATCC 25922 at 24 h (19.66 ± 0.58 mm) and 48 h (19.88 ± 1.00 mm), in contrast to the other culture broths. Therefore, TSB broth was selected as the best broth for the growth and production of antimicrobial metabolites by the *Bacillus* sp. BS4 strain (Table 2).

#### 2.3.2. Effect of Agitation, pH, and Temperature on Antimicrobial Metabolite Production

The antimicrobial activity of the BS4 strain was evident across the pH range of 5 to 9 (Figure 2). Notably, the highest antimicrobial activity was observed at both pH 7 (18.67 ± 0.58 mm) and pH 8 (19.00 ± 1.00 mm), with no significant difference between them. Consequently, pH 7 was selected as the best level for subsequent experiments.

The influence of temperature on the antimicrobial activity of the BS4 strain was examined, showing larger inhibition zones at both 28 °C (19.67 ± 0.58 mm) and 30 °C (19.83 ± 0.76 mm) with no significant difference (Figure 2). However, as the temperature further increased, antimicrobial activity decreased. Hence, 28 °C was identified as the best temperature for subsequent experiments.

Similarly, the effect of agitation on the production of antimicrobial metabolites by the BS4 strain was investigated (Figure 2). Comparable inhibition zones were observed at 150 rpm (20.33 ± 0.58 mm) and 180 rpm (20.67 ± 0.58 mm), with no significant differences. Regardless of the agitation speed, antimicrobial activity production remained largely consistent. Thus, 150 rpm was chosen as the best agitation speed for further experiments.

#### 2.3.3. Effects of Carbon, Nitrogen, and Mineral Salts Sources on Metabolite Production

The effect of the carbon source on the production of antimicrobial metabolites by the BS4 strain was evident. Results revealed that starch polysaccharide yielded the highest antimicrobial activity, with a maximum inhibition zone of 16.33 ± 1.15 mm, followed closely by glucose, which showed an inhibition zone of 16.00 ± 1.00 (Figure 3a,b). Consequently, starch was chosen as the carbon source for subsequent tests.

Similarly, a range of nitrogen sources was evaluated. Tryptone exhibited the highest antimicrobial activity, with an inhibition zone of 17.17 ± 0.76 mm, followed by yeast extract with 16.83 ± 1.76 mm. However, nitrogen sources such as ammonium sulfate and urea showed limited suitability for producing antimicrobial metabolites by the BS4 strain (Figure 4a,b). Consequently, tryptone was chosen as the nitrogen source for subsequent experiments.

The impact of mineral salt sources on antimicrobial metabolite production was assessed (Figure 5a,b). The findings revealed that magnesium sulfate (MgSO_4_) at 0.2 g/L exhibited the most significant antimicrobial activity, yielding an inhibition zone of 17.50 ± 0.50 mm, followed by calcium chloride (CaCl_2_) at 1 g/L, with a zone of 17.33 ± 1.53 mm. Therefore, magnesium sulfate at 0.2 g/L was selected as the mineral salt source for subsequent experiments.

The formulation of the modified medium (MOD) aimed to enhance antimicrobial activity was centered on starch (2.5 g/L), tryptone (20 g/L), and magnesium sulfate (2 g/L).

### 2.4. Evaluation of the Production of Biomass and Antimicrobial Metabolites Using Best Culture Conditions

The growth kinetics of the BS4 strain were compared between TSB broth and MOD broth (modified broth in Section 4.5.2 and Section 4.5.3). The logarithmic phase was observed between 3 and 9 h in both broths. Subsequently, the stationary phase began after 12 h, a period known in the literature for the highest production of antimicrobial metabolites (secondary metabolites) (Figure 6).

Antimicrobial metabolite production initiated at 6 h with MOD broth and at 9 h with TSB broth. By 9 and 12 h, the antimicrobial activity in MOD broth exhibited significantly different values, with inhibition zones of 17.50 ± 0.87 mm and 18.83 ± 1.04 mm, respectively (Figure 7). Subsequently, antimicrobial activity continued to rise over time, reaching its peak at 24 h with an inhibition zone of 19.33 ± 1.15 mm.

### 2.5. Effect of the Antimicrobial Activity of BS4 Strain by Co-Culture Method

The co-culture method was also used as another strategy to improve the antimicrobial activity of the *Bacillus* sp. BS4 strain.

Co-culture experiments were performed in both TSB and MOD broth. The inhibition zones observed through co-culture were 19.75 ± 0.96 mm at 24 h and 20.5 ± 1.29 mm at 48 h (Table 3), surpassing those observed without co-culture (Table 2).

However, the largest inhibition zone recorded was 21.85 ± 1.03 mm with MOD broth at 24 h. Additionally, co-culture was performed using heat-inactivated *E. coli* ATCC 25922 cells, resulting in less prominent inhibition zones compared to those achieved with live cells (Table 3).

### 2.6. Characterization of the Properties of the Cell-Free Supernatant of the BS4 Strain

#### 2.6.1. Evaluation of the Antimicrobial Spectrum of Selected Strain

The cell-free supernatant obtained from cultures using TSB and MOD broth was assessed. Both supernatants exhibited broad-spectrum antimicrobial activity, particularly affecting Gram-negative bacteria, which showed heightened sensitivity to the antimicrobial effects (Table 4).

Additionally, antimicrobial activity was observed against yeasts such as *C. albicans* ATCC 14053 and *C. tropicalis* ATCC 1369, a feature not observed with the cell-free supernatant obtained from TSB broth.

#### 2.6.2. Evaluation of Hemolytic and Antioxidant Activity

The entire fraction of cell-free supernatant (FC) and the purified fraction (F3K) did not exhibit hemolytic activity (Figure 8b). This suggests that the supernatant and the fraction obtained in this study are not toxic to blood cells.

Various fractions or crude extracts isolated from microorganisms have been reported to possess antimicrobial and antioxidant properties [13]. Our entire fraction of cell-free supernatant (FC) demonstrated significant free radical scavenging activity against ABTS^+^·, whereas fractions below 3 kDa did not exhibit such activity (Figure 8a). This suggests the presence of molecules with molecular weights exceeding 3 kDa that possess antimicrobial and antioxidant properties.

#### 2.6.3. Effects of Enzymes, Temperature, Surfactants, and Metal Salts on the Cell-Free Supernatant of the *Bacillus* sp. BS4 Strain

The effects of various treatments on the cell-free supernatant were examined using the well-diffusion method against *Escherichia coli* ATCC 25922. Treatment with proteinase K resulted in a partial reduction in antimicrobial activity compared to the untreated control (Appendix A). Literature evidence suggests that the treatment of supernatants with proteolytic enzymes can alter antibacterial compounds derived from proteins. Our findings align with these observations, indicating that the antimicrobial activity reduction could be attributed to peptide-derived molecules.

### 2.7. Determination of the Minimum Inhibitory Concentration (MIC)

The cell-free supernatant of the BS4 strain was filtered, lyophilized, and then resuspended in sterile physiological saline for the assessment of its antimicrobial activity through plate dilution (MIC) against Gram-negative bacteria (Figure 9). At a concentration of 7 mg/mL, it effectively inhibited the growth of both *E. coli* ATCC 25922 and *S. enterica.*

### 2.8. Phylogenetic Tree

The phylogenetic tree, constructed using the 16S rRNA gene sequences of reference strains from the *Bacillus* genus, indicates that strains BS3 and BS4 cluster together with *B. amyloliquefaciens* ATCC 23842 (EU689157) and *B. velezensis* JS25R (MF280167) (Figure 10). The 16S rRNA sequences of strains BS3 and BS4 have been deposited in GenBank under accession numbers PP396157 and PP396158, respectively.

## 3. Discussion

This study investigated the antimicrobial activity of two microbial groups: yeasts and bacteria. Among the yeasts tested, none of the 30 strains exhibited antagonistic activity against *E. coli* ATCC 25922 (Appendix A). It is likely that these yeast strains did not demonstrate zones of inhibition due to their origin. Being isolated from high Andean lagoons contaminated with mine tailings, they likely had minimal or no interaction with *E. coli*, which could have been a crucial factor in the absence of antagonistic metabolite production. 

Similar observations were reported by Acuña et al., 2010 [14], where antagonistic activity was only observed when there was direct interaction between these yeast strains and the indicator bacteria. Conversely, no inhibition zone was evident when testing the cell-free supernatant alone. This underscores the dynamic netresearch of intra- and inter-species interactions existing among microbes in their natural environment [12].

Yeasts are renowned for their intricate interactions and competitive dynamics with various cohabiting organisms, notably bacteria [15,16]. In this ecological interplay, bacteria often release a diverse array of compounds, including enzymes and fatty acids. In response, yeasts accumulate microbial stressors such as acetic acid, ethanol, and volatile antimicrobial substances [17].

In the study by Zhou et al., 2018, they explored the nutritional competition between yeasts and bacteria. They cultivated 18 yeast strains from the Saccharomycetaceae family alongside indicator bacteria for a minimum of 130 passages. Results revealed that the karyotype of certain yeasts underwent changes over successive passages, with 8 out of the 18 yeasts exhibiting genomic alterations, leading to the emergence of antibacterial activity not present in the ancestral strains. These mutant yeasts displayed enhanced ethanol production, a broader utilization of carbon sources, and increased stress tolerance. This interaction with other microorganisms could serve as a strategy to stimulate the production of secondary metabolites, a phenomenon applicable to yeasts [18]. Such interactions may mimic natural ecological settings or establish artificial communities conducive to studying the induction of secondary metabolites, particularly for the discovery of novel bioactive compounds [19,20]. Consequently, we advocate for the use of co-culture techniques to foster the production of antimicrobial metabolites by yeasts.

Conversely, the *Bacillus* genus is recognized for its production of diverse antagonistic secondary metabolites, including antibiotics, antifungals, and insecticides [21]. In this research, a total of thirty-two *Bacillus* isolates were examined, among which four strains—BS3, BS4, BS17, and BS21—displayed inhibition zones against *E. coli* ATCC 25922 (Appendix A). Remarkably, the BS4 strain exhibited larger inhibition zones compared to the other strains (Table 1).

The study by Sci et al.,2008 identified strains with notable efficacy against Gram-positive bacteria, such as *S. aureus* TISTR 517, while exhibiting no activity against Gram-negative bacteria like *E. coli* TISTR 887. They suggested the implementation of alternative evaluation methods for Gram-negative bacteria, such as cross-streak or microbial culture, instead of relying solely on free-cell supernatant [22]. In contrast, Ramachandran et al., 2015 [23] identified an antagonistic strain, *Bacillus subtilis* URID 12.1, which displayed antimicrobial activity against multidrug-resistant strains of *Staphylococcus aureus*, *S. epidermidis*, *Streptococcus pyogenes*, and *Enterococcus faecalis*.

Similarly, Yilmaz, Soran, and Beyatli, 2006 evaluated *Bacillus brevis* strains exhibiting antagonistic activity against *S. aureus* ATCC 25923 while showing no inhibitory effect against Gram-negative bacteria such as *E. coli* NRRL B-704, *Yersinia enterocolitica* ATCC 1501, and *Pseudomonas aeruginosa* ATCC 27853 [24]. In accordance with our findings, our strain demonstrated high antimicrobial activity against both Gram-positive and Gram-negative bacteria as well as yeasts, underscoring its broad-spectrum action (Table 4).

In this study, the antimicrobial production of strains was observed after 24 h, 48 h, and 72 h of incubation time in TSB broth using both CB and CFS, with no significant differences (*p* < 0.05) in their antimicrobial activity against *E. coli* ATCC 25922 (Table 1). This suggests that 24 h of incubation is sufficient for antimicrobial metabolite production, aligning with the results of Barale, Ghane, and Sonawane, 2022 [25]. Furthermore, Usta and Demirkan, 2013 evaluated the influence of time on antagonist activity (24 h, 48 h, 72 h, and 96 h of incubation), finding that the maximum antagonist activity was attained at 72 h [26]. They also highlighted that the overall antimicrobial activity is contingent upon the microbial strain and the metabolic pathways engaged. Nevertheless, for *Bacillus* species, antimicrobial activity typically manifests between 24 and 72 h of incubation.

In the last two decades, nearly 1000 antimicrobial compounds have been discovered; however, the rediscovery of known compounds remains a significant bottleneck in this field [27]. Therefore, various approaches have been explored for the discovery of novel molecules. The variation in physicochemical parameters plays a crucial role in facilitating the discovery of new antimicrobial molecules by creating a conducive environment for the expression of cryptic genes and, consequently, the production of novel bioactive molecules [10]. For instance, Zeeck and colleagues identified more than ten additional metabolites produced by the fungus *Aspergillus ochraceus* through modifications in parameters such as salinity and temperature [28]. Hence, in this study, we employed the OSMAC strategy using the one-factor-at-a-time method to elucidate the behavior of our strain and optimize the production of antimicrobial metabolites.

In our investigation, we assessed the influence of three different culture broths—TSB, LB, and YPG—on both biomass and antimicrobial metabolites production. TSB broth was selected because high amounts of biomass were produced, and the evident inhibition zones had an inhibition zone of 19.66 ± 0.58 mm at 24 h compared to the LB and YPG media (Table 2). Subsequently, we tested different physicochemical (pH, temperature, and agitation) and nutrient parameters to increase the production of antimicrobial metabolites of the BS4 strain (Figure 2, Figure 3, Figure 4 and Figure 5).

The influence of carbon and nitrogen sources was significant. Carbon sources play a crucial role in providing energy and precursors necessary for biomass synthesis and the production of secondary metabolites [29]. Rapidly assimilated carbon sources may lead to adverse effects, such as catabolic repression or the “glucose effect” [30], which can be mitigated by utilizing complex carbon sources such as polysaccharides or dextrins [31]. In this study, the antimicrobial activity was significantly different when starch was used, resulting in inhibition zones of 16.33 ± 1.15 mm, representing a notable increase compared to the inhibition zone obtained with the control medium lacking a carbon source, which measured 14.33 ± 0.58 mm (Figure 3a,b). Furthermore, in the study by Messis et al., 2014, starch as a carbon source was found to enhance the production of antifungals by the *Streptomyces* sp. TKJ2 strain [32].

Similarly, the nitrogen source acts as a precursor for the synthesis of amino acids, nitrogenous compounds, proteins, DNA, and RNA. Moreover, it supports bacterial growth and facilitates the production of secondary metabolites, with amino acids playing a structural role in antimicrobial peptides [33]. Our study identified tryptone as the most effective nitrogen source, resulting in an inhibition zone of 17.17 ± 0.76 mm (Figure 4a,b). It is noteworthy that the absence of a nitrogen source in the base medium led to inhibited antimicrobial activity, underscoring the essential role of nitrogen sources in metabolite production. Comparably, Saraniya and Jeevaratnam, 2014, optimized nutritional factors to enhance the production of bacteriocins in *Lactobacillus pentosus* SJ65, where they found that soy tryptone and peptone are excellent sources of nitrogen [34].

Additionally, co-cultivation represents a promising strategy for this purpose. In our study, we examined the co-culture of the antagonist BS4 strain with the reference strain *E. coli* ATCC 25922 (Table 3). Co-cultivation resulted in increased antimicrobial activity, observed with both live and heat-inactivated *E. coli* cells, albeit to a lesser extent with the heat-inactivated cells. A similar outcome was noted in the study by Benitez et al., 2011, wherein the *Bacillus amyloliquefaciens* LBM 5006 strain exhibited enhanced antimicrobial activity when cultured alongside thermally inactivated cells, live cells, and cell debris following *E. coli* cell fractionation. This suggests that the stimulating factor is associated with cells, thereby promoting the synthesis of antimicrobial peptides by *B. amyloliquefaciens* LBM 5006 [35]. Co-culture enhances the production of antimicrobial metabolites by simulating the natural interactions among microorganisms and the accompanying physical and chemical transformations they undergo. These interactions may be mediated by diffusible molecules or necessitate direct cell–cell contact, potentially influencing cellular processes and leading to the expression of novel bioactive molecules, thereby enhancing competitiveness [10].

The strains BS3 and BS4 exhibit a close genetic relationship at the 16S rRNA gene level with reference strains *B. amyloliquefaciens* ATCC 23842 (EU689157) and *B. velezensis* JS25R (MF280167), both belonging to the *Bacillus subtilis* group (Figure 10) [36]. *B. amyloliquefaciens* strains are renowned for their efficacy as biological control agents to the extent that they are incorporated into commercial inoculants for agricultural biological control [37,38]. These strains have been reported to produce various antibiotics and antibacterial metabolites, including cyclic lipopeptides such as surfactin, bacillomycin, fengycin, the iron siderophore bacillibactin, amylolysin, and subtilosin A, among others [39]. Similarly, *B. velezensis* strains are recognized as biocontrol agents against plant pathogens and producers of lipopeptides and polyketides compounds [40,41,42].

In this study, we demonstrated that the cell-free supernatant exhibited radical scavenging activity for ABTS^+^ (Figure 8a) through color change. These findings suggest that the cell-free supernatant may contain molecules with antioxidant and antimicrobial capacity, as well as the absence of hemolytic activity through erythrocytes (Figure 8b). These experiments confirmed that the cell-free supernatant possesses both antimicrobial and antioxidant activity and is not hemolytic. Based on the results, the strain BS4 under study is a candidate for further investigation at the genomic, transcriptomic, and antimicrobial metabolite characterization levels to understand the mechanisms underlying antimicrobial activity and optimize production comprehensively.

## 4. Materials and Methods

### 4.1. Materials

Culture media (Tryptic Soy Broth, Tryptic Soy Agar, Peptone, Yeast Extract, Tryptone, Glucose, and Bacteriological Agar) were purchased from HiMedia (Mumbai, India); ABTS from Sigma Aldrich Co. (St. Louis, MO, USA); and ciprofloxacin disks from Bioanalyse^®^ (Ankara, Turkey).

### 4.2. Strain and Growth Conditions

For this study, twenty-seven strains of yeasts were isolated from surface water contaminated with mine tailings in the Central Sierra of Peru (Junin and Pasco Region), three strains of yeasts were isolated from fertile soil where cocoa (*Theobroma cacao* L.) grows in the province of Pichanaqui (Junin), and thirty-two *Bacillus* strains were obtained from the collection of the Laboratory of Environmental Microbiology and Biotechnology at the Faculty of Biological Sciences, Universidad Nacional Mayor de San Marcos. The reference strains included *Staphylococcus aureus* ATCC 25923, *Staphylococcus aureus* ATCC 6538, *Staphylococcus epidermidis* ATCC 1228, *Staphylococcus saprophyticus* ATCC 15305, *Citrobacter freundii* ATCC 8090, *Escherichia coli* ATCC 25922, *Proteus vulgaris* ATCC 49132, *Enterobacter aerogenes*, *Salmonella enterica*, *Shigella flexneri* ATCC 12022, *Yersinia enterocolitica* ATCC 23715, *Pseudomonas aeruginosa*, *Serratia marcescens* ATCC 14756, *Aeromonas* sp., *Candida albicans* ATCC 14053, and *Candida tropicalis* ATCC 1369, as well as nosocomial strains of *Klebsiella pneumoniae*, *Escherichia coli*, *Pseudomonas aeruginosa*, and *Acinetobacter baumannii.* Bacteria and yeasts were reactivated in Tryptic Soy Broth at pH 7.2 and Yeast Peptone Glucose broth at pH 5.0, respectively.

### 4.3. Screening of Antagonistic Yeast 

The antagonistic activity of yeasts under study against *Escherichia coli* ATCC 25922 was determined by the ‘Spot technique’ method [14,43]. First, 100 μL of *E. coli* ATCC 25922 suspension (~1.5 × 10^8^ cells/mL) were spread using a sterile swab onto plates containing YGSA solid medium (5 g/L yeast extract, 20 g/L glucose, 10 g/L tryptone, 10 g/L NaCl, and 15 g/L agar) adjusted to pH 5.0. Ten microliters of the yeast suspensions per spot were then inoculated onto the agar surface, with a total of five yeast suspensions evenly distributed on the plate. The cultures were incubated at 28 °C for 72 h. The negative control consisted of the culture broth, while the positive control was the antibiotic ciprofloxacin (5 μg).

### 4.4. Screening of Antagonistic Bacteria

The antagonistic activity against *Escherichia coli* ATCC 25922 was determined by the agar well-diffusion method [24], employing both bacterial culture (BC) and cell-free supernatant (CFS). To obtain the CFS, cultures from 18 to 24 h were centrifuged at 5000 rpm for 30 min, and the resulting supernatants were filter-sterilized with a 0.22 μm filter. Subsequently, 100 μL of either BC or CFS were dispensed into each well. The plates were incubated at 37 °C for 24 h. The negative control consisted of TSB broth, while the positive inhibition control was the antibiotic ciprofloxacin (5 μg).

### 4.5. Effect of Culture Conditions on Improving the Production of Antimicrobial Metabolites from the Bacillus sp. BS4 Strain

The selected strain was employed to assess the effect of culture conditions on the production of antimicrobial metabolites.

#### 4.5.1. Selection of Fermentation Medium 

Various broth cultures were utilized to determine the best medium for the growth and antimicrobial activity of the *Bacillus* sp. BS4 strain: Luria–Bertani broth (LB) (tryptone, 10 g/L; yeast extract, 5 g/L; NaCl, 10 g/L); Tryptone Soy Broth (TSB) (tryptone, 17 g/L; soybean, 3 g/L; glucose, 2.5 g/L; NaCl, 5.0 g/L; K_2_HPO_4_, 2.5 g/L); and Yeast Extract–Peptone–Glucose broth (YPG) (yeast extract, 5.0 g/L; peptone, 10 g/L; glucose, 30 g/L. All media were adjusted to a pH of 7. A total of 5 mL of the initial inoculum, with a concentration of ~3 × 10^8^ CFU/mL, was transferred to a 100 mL flask with 45 mL of culture broth and incubated at 28 °C on a rotary shaker at 120 rpm for 72 h. A sample of each medium was extracted at 24, 48, and 72 h and centrifuged at 5000 rpm for 30 min at room temperature. Subsequently, each sample was filtered using a filter with a pore size of 0.22 µm (Merck Millipore, Darmstadt, Germany) to obtain a cell-free supernatant (CFS) [44]. The antagonism activity was assessed by the agar well-diffusion method [24]. A 5 µg disk of the antibiotic ciprofloxacin and the culture medium served as positive and negative controls, respectively. The experiments were performed in triplicate.

Bacterial growth was monitored by optical density at 600 nm, followed by plating appropriate dilutions onto TSA agar plates. After an incubation period of 24 h at 28 °C, the resulting colonies were enumerated. This experimental protocol was replicated in triplicate and expressed in Log10 (CFU/mL).

#### 4.5.2. Effect of Agitation, pH, and Temperature on Antimicrobial Metabolite Production Using the One-Factor-at-a-Time Method

The temperature, pH, and agitation effects were assessed in the selected broth [45]. Initially, a 5 mL inoculum (3 × 10^8^ CFU/mL) of the strain was transferred to a 100 mL flask containing 45 mL of TSB broth. The parameters investigated included temperature (28 °C, 30 °C, 35 °C, and 37 °C), pH levels (4, 5, 6, 7, 8, 9, and 10), and agitation rates (90, 120, 150, and 180 rpm). Antimicrobial activity was evaluated using the previously described assay. The one-factor-at-a-time method was employed, wherein each factor was sequentially altered.

#### 4.5.3. Effect of Carbon, Nitrogen, and Mineral Salt Sources on Metabolite Production Using the One-Factor-at-A-Time Method

For this experiment, we adopted the methodology outlined by Sa-Uth et al., 2018 based on the one-factor-at-a-time approach to determine the best sources of nitrogen, carbon, and mineral salts to increase the antagonist activity of the *Bacillus* sp. BS4 strain [44].

Different sources of nitrogen (bacteriological peptone, tryptone, yeast extract, meat extract, ammonium sulfate, and urea), carbon (glucose, fructose, lactose, sucrose, starch, and glycerol), and mineral salts (sodium chloride 5 g/L, dipotassium phosphate 2 g/L, magnesium sulfate 0.2 g/L and 1 g/L, zinc sulfate 0.2 g/L, sulfate manganese 0.05 g/L, and calcium chloride 0.88 g/L) [46] were used individually to replace the corresponding sources in the original medium.

The antagonistic activity was determined by the agar well-diffusion method against *Escherichia coli* ATCC 25922. The data obtained were evaluated by analysis of variance (ANOVA).

### 4.6. Evaluation of the Production of Biomass and Antimicrobial Metabolites Using Favorable Culture Conditions

The inoculum of the BS4 strain was incubated for 18–24 h under the best culture conditions. Subsequently, an inoculum of 5 mL of this culture was transferred to a 100 mL Erlenmeyer flask containing 45 mL of TSB and MOD broth [47].

The culture was incubated at 28 °C and agitated at 150 rpm for 24 h. Samples were collected at 3 h intervals to recover cell-free supernatant (CFS), as described above. Optical density at 600 nm was monitored, followed by plating appropriate dilutions on Trypticase soy agar (TSA) plates. The plates were incubated at 28 °C for 24 h, and the resulting colonies were enumerated. The experiment was conducted in triplicate, and the values were expressed in Log10 (CFU/mL) [23]. Subsequently, biomass values and inhibition zones obtained with both TSB and MOD broth culture broth were compared.

### 4.7. Effect of the Antimicrobial Activity of the Bacillus sp BS4 Strain by Co-Culture Method

The co-culture was selected as another alternative to enhance the antimicrobial activity of the BS4 strain against *E.coli* ATCC 25922, according to the methodology of Benitez et al., 2011, with modifications [35].

A latent phase culture of *E. coli* ATCC 25922 was prepared to an optical density (OD) of 0.5, as per the McFarland scale (~1.5 × 10^8^ cells/mL). Subsequently, 1 milliliter of this culture was centrifuged at 13,000 rpm for 15 min. The resulting pellet was washed three times with 0.9% saline solution and resuspended in 1 mL of broth culture. This resuspended culture was then added to a flask containing 44 mL of both Tryptic Soy Broth (TSB) and Modified (MOD) broth, respectively. Additionally, 5 mL of a 24 h latent phase culture of the BS4 strain was introduced into the same flask. The co-culture was then incubated at 28 °C for 48 h on an orbital shaker set at 150 rpm. Furthermore, a co-culture test was conducted in the presence of thermally inactivated *E. coli* ATCC 25922 cells, which were exposed to 80 °C for 30 min under the same incubation conditions. As a control, an independent culture of the BS4 strain was maintained.

Each co-culture was centrifuged and filtered to obtain supernatants. Each one was tested by agar well-diffusion method against the same reference bacteria (*E. coli* ATCC 25922), and the inhibition zones obtained were measured. 

### 4.8. Characterization of the Antimicrobial Properties of the Cell-Free Supernatant of the Bacillus sp. BS4 Strain

#### 4.8.1. Evaluation of the Antimicrobial Spectrum of the Selected Strain

The BS4 strain was selected to evaluate its activity against several bacteria and yeasts. Antagonistic activity was measured using the same method previously described. 

#### 4.8.2. Concentrated Fractions of the Supernatant

The cell-free supernatant was dialyzed against distilled water using a dialysis bag of 1000 Da to eliminate traces of salts and sugars from the culture medium. The content of the culture was recovered from a dialysis bag and purified with an Amicon ultra centrifugal filter (Merck Millipore, Darmstadt, Germany) with a cut-off from 3 kDa. Finally, the eluate was again concentrated with a rotary evaporator [48].

#### 4.8.3. Evaluation of Hemolytic and Antioxidant Activity

The hemolytic activity of the fractions was assessed using the methodology described by Oddo and Hansen, 2017 [49]. A 0.5% concentration of red blood cells in PBS was incubated at 37 °C for 3 h with varying dilutions of FC (entire fraction of cell-free supernatant) and F3K (3 kDa fraction) in 0.2 mL polypropylene tube strips, allowing gravity to facilitate the formation of precipitate. Subsequently, the tubes were centrifuged to obtain supernatants, which were then read at 405 nm. The hemolysis results were compared with positive (SDS 0.25%) and negative (PBS 1X) controls.

The antioxidant activity was determined using the method described by Perez et al., 2019 [48], wherein 20 µL of FC (entire fraction of cell-free supernatant) and F3K (3 kDa fraction) were incubated with ABTS (2,2′-azino-bis(3-ethylbenzothiazoline-6-sulfonic acid)), followed by measurement of absorbance at 734 nm.

#### 4.8.4. Effects of Enzymes, Temperature, Surfactants, and Metal Salts on the Cell-Free Supernatant of the *Bacillus* sp. BS4 Strain

The effects of enzymes, temperature, surfactants, and metal salts on the CFS were investigated. The effect of the different treatments of the cell-free supernatant was determined by the well-diffusion method against *Escherichia coli* ATCC 25922. The untreated CFS was used as a control.

Enzyme Effect

The impact of enzymes on the cell-free supernatant (CFS) was assessed using proteinase K enzyme at concentrations of 2 and 10 mg/mL. The CFS was incubated at 37 °C for 1 h in the presence of the enzyme. Subsequently, the antagonism assay was performed [50].

Stability of Antimicrobial Activity

The thermal stability of the CFS was determined by exposing it to different incubation temperatures: 30 °C, 60 °C and 100 °C for 30 min and 121 °C for 20 min (autoclaving) [51].

Effect of Surfactants and Other Chemicals

The effect of surfactants and other chemicals was determined using SDS, Tween 80, and urea at a final concentration of 1% (*v*/*v*) and EDTA at 2, 5, and 50 mM. These substances were mixed with the CFS and incubated at 37 °C for 5 h [51].

Effect of Metallic Salts

The effect of the metal salts was evidenced by mixing the CFS with the following metal salts: MgSO_4_, ZnSO_4_, FeSO_4_, and CaCl_2_ at a final concentration of 1 mg/mL. Both were incubated at 37 °C for one hour. The untreated supernatant and metal salt solutions were used as positive and negative controls, respectively [50].

### 4.9. Determination of the Minimum Inhibitory Concentration (MIC)

The minimum inhibitory concentration (MIC) was evaluated using the 96-well plate microdilution method following the protocol described by Wiegand et al., 2008 [52]. The experiment was performed in Luria–Bertani (LB) broth with a final inoculum of 10^4^ CFU/mL, adjusted from a 0.5 McFarland bacterial suspension. Different concentrations of the lyophilized supernatants (in μg/mL) obtained from co-culture supernatant were evaluated in 100 μL of Luria–Bertani broth. Subsequently, the plates were incubated at 37 °C for 18 h, and the optical density at 600 nm was measured using a spectrophotometer. All tests were conducted in triplicate. Chloramphenicol (0.04 mg/L) was employed as the positive control, while the bacterial culture served as the negative control.

### 4.10. Genomic DNA Extraction

Bacterial strains were cultured in TSB broth and incubated at 28 °C on a rotary shaker at 150 rpm for 18–24 h. Each bacterial biomass was recovered by centrifugation and adjusted for total DNA extraction according to the instructions of the Wizard^®^ genomic DNA purification Kit (Promega, Madison, WI, USA).

### 4.11. 16S rRNA Gene Amplification and Nanopore Sequencing

The LongAmp Taq 2X Master Mix kit (New England Biolabs, Ipswich, MA, USA) was employed to amplify approximately 1500 bp of the 16S rRNA gene using primers 27F and 1492R, following cycling conditions outlined by Navarrete et al., 2010 [53]. The PCR products were purified according to the instructions provided by the NucleoTraP^®^ CR Kit (Macherey-Nagel, Allentown, PA, USA). Sequencing of the PCR products was carried out using the Rapid Barcoding Kit SQK-RBK004 (Oxford Nanopore Technologies, Oxford, UK) within an R9.4.1 flow cell.

Fastq files obtained from base calling were subsequently uploaded to the NanoGalaxy web server (https://nanopore.usegalaxy.eu/, accessed on 7 October 2023) for analysis and consensus sequence derivation [54]. Briefly, the research flow starts with Filtlong for filtering sequences of appropriate length and high quality (score ≥ 8), Porechop for removal of barcode sequences, Minimap2 for alignment to a reference sequence, and finally, Medaka consensus for generating a consensus sequence.

### 4.12. Phylogenetic Analysis

All 16S rRNA sequences, including those obtained in this study, were aligned using the MUSCLE algorithm included in MEGA software version 11 [55]. Cut 1381 base pairs of aligned 16S rRNA sequences were analyzed using the neighbor-joining statistical method with 1000 bootstrap replications in MEGA 11 software. Reference sequences were retrieved from GenBank (https://www.ncbi.nlm.nih.gov/genbank/, accessed on 10 November 2023).

### 4.13. Statistical Analyses

All experiments were performed with three biological replicates. The data were expressed as the mean ± standard error of the mean (SEM) and analyzed using one-way ANOVA. Group differences were assessed using the least significant difference (LSD) test via GraphPad Prsim version 8.0 software (GraphPad, San Diego, CA, USA). Statistical significance was set at *p* < 0.05.

## 5. Conclusions

This study found four strains of the genus *Bacillus* capable of producing antimicrobial metabolites. Among these, the BS3 and BS4 strains were classified as *Bacillus velezensis* and *Bacillus amyloliquefaciens*, respectively. Following screening procedures, the BS4 strain was chosen for subsequent investigations.

The effect of physicochemical parameters and culture medium composition was evaluated using the one-factor-at-a-time method, revealing significant improvements in metabolite production. The best conditions included a pH of 7, a temperature of 28 °C, agitation at 150 rpm, starch (2.5 g/L) as the carbon source, tryptone (20 g/L) as the nitrogen source, and magnesium sulfate (0.2 g/L) as the mineral salt source. Additionally, co-cultivation proved to be an effective strategy for enhancing metabolite production, resulting in larger inhibition zones against *E. coli* ATCC 25922 compared to standard culture conditions.

The antimicrobial metabolites synthesized by this strain exhibited broad-spectrum activity, particularly potent against Gram-negative bacteria such as *Salmonella enterica*, *Klebsiella pneumoniae*, *Shigella flexneri*, *Enterobacter aerogenes*, *Proteus vulgaris*, *Yersinia enterocolitica*, *Serratia marcescens*, *Aeromonas* sp., *Pseudomonas aeruginosa*, and yeasts such as *Candida albicans* and *C. tropicalis*. Furthermore, these metabolites displayed antioxidant and non-hemolytic properties, indicating their potential for novel molecule discovery. The preliminary characterization suggested a probable proteinaceous nature of these metabolites, warranting further chemical analysis to elucidate their composition.

## Figures and Tables

**Figure 1 antibiotics-13-00304-f001:**
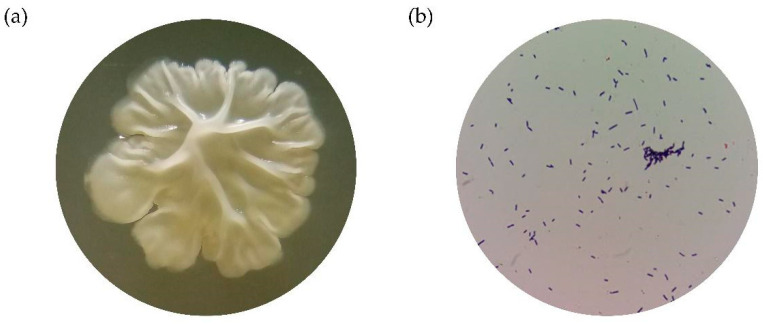
Phenotypic characterization of the *Bacillus* sp. BS4 strain: (**a**) strain growth on TSA agar at 24 h reveals irregular size and shape, with a raised, butyrous appearance and mucoid consistency, observed using a stereomicroscope with 10× magnification; and (**b**) Gram stain (100× magnification).

**Figure 2 antibiotics-13-00304-f002:**
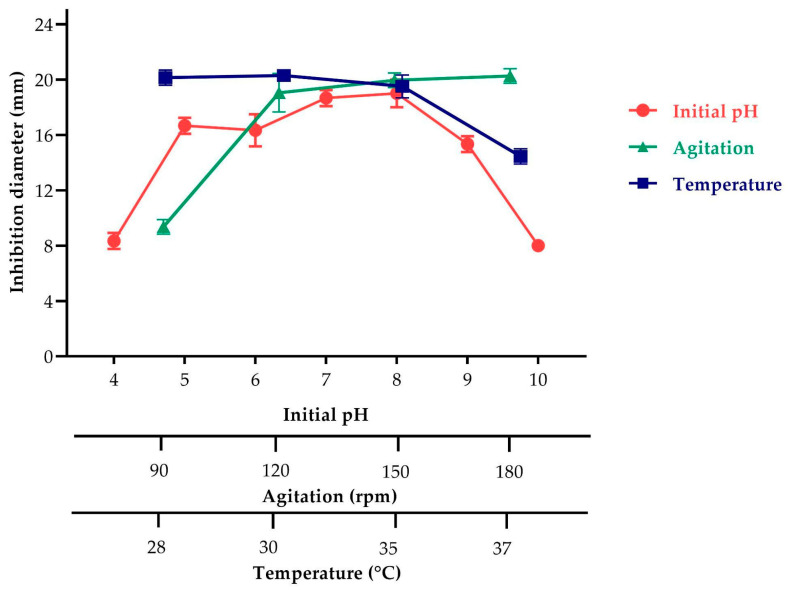
Effect of agitation, pH, and temperature on antimicrobial metabolite production of the BS4 strain against *E. coli* ATCC 25922.

**Figure 3 antibiotics-13-00304-f003:**
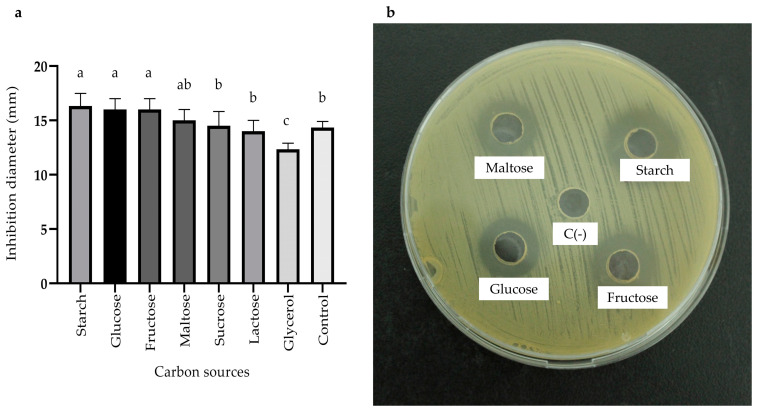
Effects of various carbon sources on antimicrobial activity of the *Bacillus* sp. BS4 strain. (**a**) Bar graph represents the effect of carbon sources and (**b**) Inhibition zones of CFS. Distinct letters represent statistically significant differences between treatments based on the LSD test (*p* < 0.05).

**Figure 4 antibiotics-13-00304-f004:**
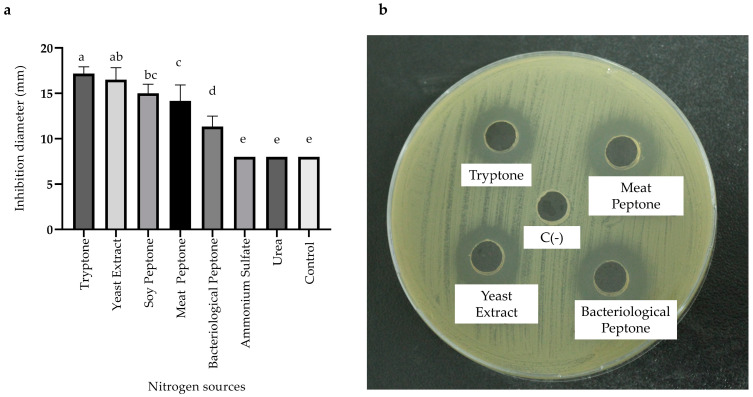
Effects of various nitrogen sources on antimicrobial activity of the *Bacillus* sp. BS4 strain. (**a**) Bar graph represents the effect of nitrogen sources and (**b**) Inhibition zones of CFS Distinct letters represent statistically significant differences between treatments based on the LSD test (*p* < 0.05).

**Figure 5 antibiotics-13-00304-f005:**
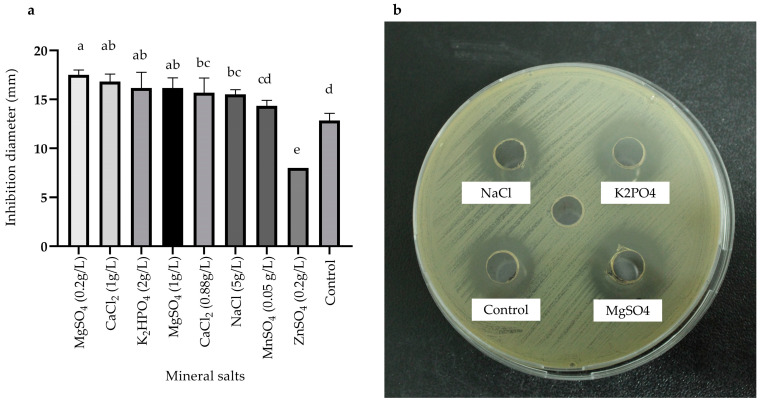
Effects of various mineral salts on antimicrobial activity of the *Bacillus* sp. BS4 strain. (**a**) Bar graph represents the effect of mineral salts sources and (**b**) Inhibition zones of CFS. Distinct letters represent statistically significant differences between treatments based on the LSD test (*p* < 0.05).

**Figure 6 antibiotics-13-00304-f006:**
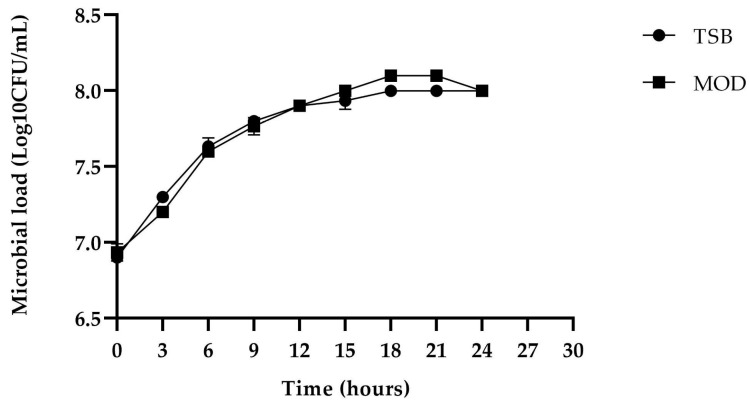
Comparison of the growth kinetics of the BS4 strain in TSB (
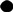
) and MOD broth (
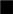
). The error bars in the figure indicate the standard deviations from three replicates.

**Figure 7 antibiotics-13-00304-f007:**
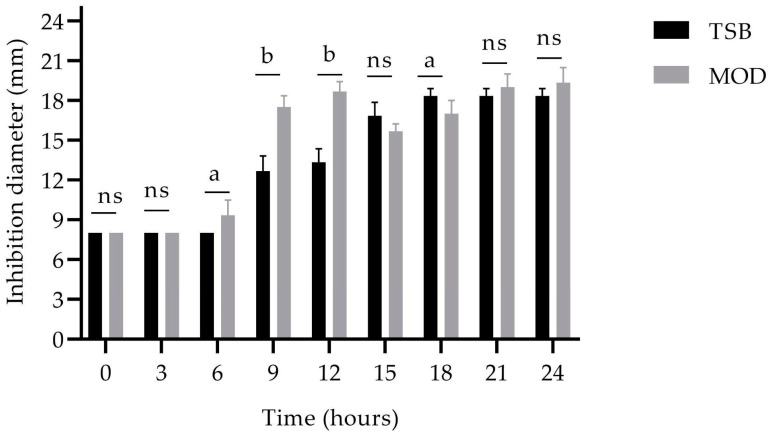
Comparisons of the antimicrobial activity of the BS4 strain in TSB broth and MOD broth. Distinct letters represent statistically significant differences between treatments based on the LSD test where ns, not significant; a, *p* < 0.05; b, *p* < 0.001.

**Figure 8 antibiotics-13-00304-f008:**
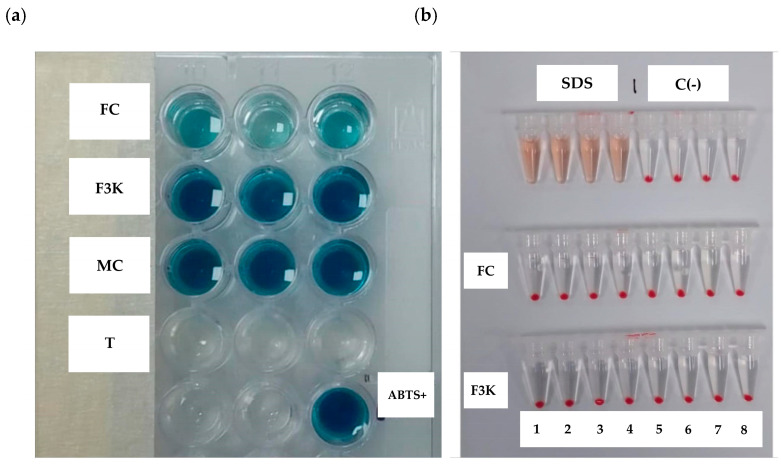
Evaluation of antioxidant (**a**) and anti-hemolytic (**b**) activity of the *Bacillus* sp. BS4 strain. FC, entire fraction of cell-free supernatant; F3K, 3 kDa fraction; MC, culture medium; T, Trolox; ABTS, solution of ABTS^+^ (2,2′-azino-bis(3-ethylbenzothiazoline-6-sulfonic acid).

**Figure 9 antibiotics-13-00304-f009:**
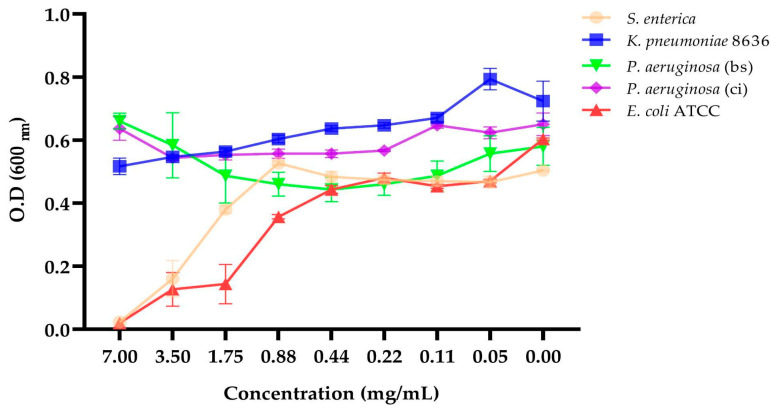
MIC of different Gram-negative bacteria in the presence of different concentrations of CFS. These data represent the mean (±SEM) of three independent experiments (bs, bronchial secretion; ci, clinical isolation).

**Figure 10 antibiotics-13-00304-f010:**
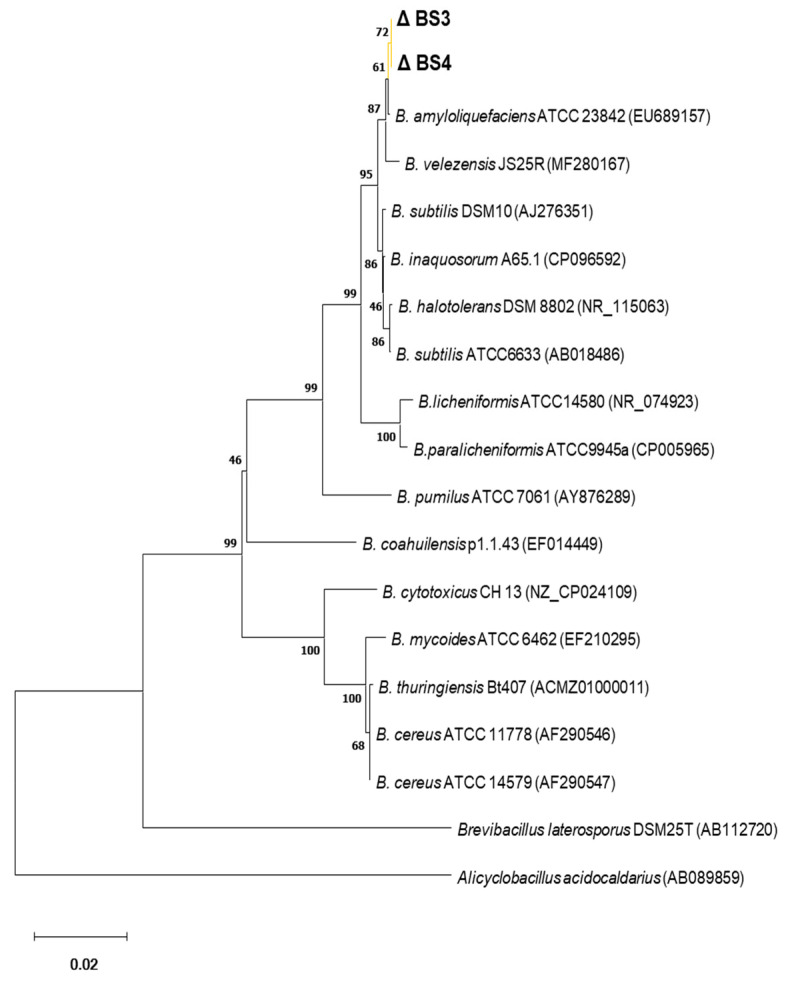
Phylogenetic relationship of strains BS3 and BS4 (Δ) with the type strains of the *Bacillus* genus. Sequences were aligned using MUSCLE (MEGA software version 11), and phylogenetic inferences were obtained using neighbor joining with 1000 bootstrap replicates in MEGA 11. *Brevibacillus laterosporus* DSM25T (AB112720) and *Alicyclobacillus acidocaldarius* (AB089859) were used as outgroups. Scale bar = 1% nucleotide sequence divergence.

**Table 1 antibiotics-13-00304-t001:** Antimicrobial activity of bacterial culture and CFS of *Bacillus* strains in different incubation times (24, 48, and 72 h) against *E. coli* ATCC 25922.

Strains	Antimicrobial Activity (mm ± SD)					
	24 h		48 h		72 h	
	BC	CFS	BC	CFS	BC	CFS
BS3	19.00 ± 1.41 ^a^	17.67 ± 0.58 ^a^	19.50 ± 1.41 ^a^	18.50 ± 1.53 ^a^	18.00 ± 0 ^b^	18.50 ± 1.00 ^b^
BS4	20.50 ± 0.70 ^a^	19.67 ± 0.58 ^a^	20.00 ± 1.41 ^b^	20.00 ± 1.00 ^b^	18.50 ± 0.71 ^a^	19.33 ± 0.58 ^a^
BS17	19.00 ± 2.83 ^a^	18.00 ± 1.80 ^a^	18.50 ± 2.12 ^c^	17.67 ± 1.15 ^c^	14.50 ± 2.12 ^a^	19.17 ± 0.76 ^a^
BS21	18.50 ± 3.54 ^b^	17.00 ± 1.73 ^b^	18.30 ± 0.70 ^c^	17.33 ± 0.58 ^c^	17.33 ± 1.00 ^d^	16.50 ± 1.00 ^d^
Control (+)	26.00 ± 1.15	26.00 ± 1.15	26.00 ± 1.15	26.00 ± 1.15	26.00 ± 1.15	26.00 ± 1.15
Control (−)	-	-	-	-	-	-

BC: Bacterial culture, CFS: cell-free supernatant. Distinct letters represent statistically significant differences between treatments based on the LSD test (*p* < 0.05).

**Table 2 antibiotics-13-00304-t002:** Effect of different media on the biomass and antimicrobial activity of *Bacillus* sp. BS4 strain against *E. coli* ATCC 25922 after cultivation in different periods of time.

Culture Medium	Biomass (Log10 (CFU/mL ± SD))			Diameter of Inhibition Zone (mm, Mean ± SD)		
	24 h	48 h	72 h	24 h	48 h	72 h
TSB	8.86 ± 0.01 ^a^	8.79 ± 0.06 ^b^	8.77 ± 0.08 ^c^	19.66 ± 1.15 ^a^	19.88 ± 1.00 ^b^	19.33 ± 1.15 ^b^
LB	8.72 ± 0.04 ^c^	8.72 ± 0.01 ^a^	8.73 ± 0.05 ^b^	15.67 ± 0.58 ^a^	14.67 ± 0.58 ^b^	14.00 ± 1.00 ^a^
YPG	8.80 ± 0 ^b^	8.70 ± 0.07 ^c^	8.71 ± 0.08 ^a^	18.00 ± 1.00 ^a^	15.33 ± 0.58 ^a^	14.67 ± 1.53 ^a^

Distinct letters represent statistically significant differences between treatments based on the LSD test (*p* < 0.05).

**Table 3 antibiotics-13-00304-t003:** Inhibition zones obtained from the co-culture method of the *Bacillus* sp. BS4 strain and *E. coli* ATCC 25922.

Treatment	Zone of Inhibition (mm)	
	24 h	48 h
TSB + *E. coli*	19.75 ± 0.96 ^b^	20.5 ± 1.29 ^b^
MOD + *E. coli*	21.85 ± 1.03 ^a^	21.12 ± 1.03 ^b^
MOD + inactivated *E. coli*	20.25 ± 0.50 ^b^	20.50 ± 0.58 ^b^

Distinct letters represent statistically significant differences between treatments based on the LSD test where a indicates not significant and b represents *p* < 0.05.

**Table 4 antibiotics-13-00304-t004:** Antimicrobial activity of the *Bacillus* sp. BS4 strain by agar well-diffusion assay using the CFS from TSB and MOD at 24 h.

Indicator Strains	TSB	MOD
*Staphylococcus aureus* ATCC 6538	N.A.	N.A.
*Staphylococcus aureus* ATCC 25923	N.A.	N.A.
*Staphylococcus epidermidis* ATCC 1228	N.A.	N.A.
*Staphylococcus saprophyticus* ATCC 15305	N.A.	16.38 ± 0.38
*Citrobacter freundii* ATCC 8090	16.00 ± 0	16.33 ± 0.58
*Escherichia coli* ATCC 25922	19.67 ± 0.58	20.67 ± 0.58
*Proteus vulgaris* ATCC 49132	19.33 ± 1.15	20.33 ± 0.58
*Enterobacter aerogenes*	17.67 ± 0.58	18.33 ± 0.58
*Salmonella enterica*	20.33 ± 0.58	23.67 ± 0.58
*Shigella flexneri* ATCC 12022	15.00 ± 0	16.00 ± 0
*Pseudomonas aeruginosa*	N.A.	N.A.
*Yersinia enterocolitica* ATCC 23715	21.67 ± 1.15	22.00 ± 1.00
*Serratia marcescens* ATCC 14756	21.67 ± 0.58	22.33 ± 0.58
*Aeromonas* sp.	29.00 ± 0.15	30.00 ± 0.15
*Candida albicans* ATCC 14053	N.A.	11.33 ± 1.15
*Candida tropicalis* ATCC 1369	N.A.	13.33 ± 1.15
Nosocomial Strains		
*Klebsiella pneumoniae* 8636 (clinic isolation)	24.50 ± 0.50	25.67 ± 0.58
*Escherichia coli* (fistula, clinic isolation)	19.00 ± 1.00	21.17 ± 0.29
*Escherichia coli* BLEE 1705 (clinic isolation)	16.00 ± 0	16.33 ± 1.15
*Escherichia coli* BLEE 78 (clinic isolation)	16.00 ± 1.00	16.00 ± 1.00
*Escherichia coli* BLEE 79 (clinic isolation)	N.A.	N.A.
*Pseudomonas aeruginosa* (urine culture)	13.33 ± 1.53	13.67 ± 0.30
*Pseudomonas aeruginosa* (bronchial secretion)	14.33 ± 0.58	N.A.
*Pseudomonas aeruginosa* (clinic isolation)	16.00 ± 0.00	15.67 ± 0.30
*Acinetobacter baumannii* (clinic isolation)	N.A.	N.A.

N.A.: no activity.

## Data Availability

Data are contained within this article and the Appendix A.

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
