# Peer review of "Antimicrobial Activity of *Bacillus amyloliquefaciens* BS4 against Gram-Negative Pathogenic Bacteria"

_antibiotics, 2024, doi:10.3390/antibiotics13040304_

Round 1
Reviewer 1 Report
Comments and Suggestions for Authors
General comment:
The paper is not novel and the results reported are too premature for publication. More work is needed to substantiate the conclusion in your manuscript.
Other comments:
Title
the term fraction is not appropriate, change it.
“isolated from Peru”, Peru is not a sample where microorganisms can be isolated; revise that, and mention the samples from which the strain was isolated.
Abstract
Provide criteria for selecting the Gram-negative strain Escherichia coli ATCC 25922 for your research and no other pathogens.
Line 14, separates the general objective of the work and the methodology
Line 22, these strains should be included in the methodology
Line 26-27,” the production is influenced by factors such as.. .” revise that sentence and add all the significant factors arising from the screening design performed in your study.
line 30, add the concentration of broth ingredients like starch, tryptone and magnesium sulfate.
Bacterial resistance cannot stand as a keyword, as resistance was not assessed in this study. remove. Add Bacillus sp. BS4
Introduction
line 56, provide a paragraph that might justify why yeast strains should be used in this study.
Line 85, there are some interesting antimicrobial metabolites produced by microorganisms that remain attached to cell wall and need specific treatment to be detached. Why these metabolites were not considered in this study?
Results
Line 101, these tests might be insufficient to assign the isolate to Bacillus genus
Table 1, perform LSD to compare means. This will be the scientific base to select BS4 as the main important strain. also, mention E. coli in the table title.
Line 119, as there was no significant difference you cannot select TSB
Table 2, add LSD superscript to compare means according to time and strains
What happens above 72 h? it will bring more information on the growth phase and ease the selection of optimal conditions. Experiment should have been performed at 96 h.
Line 177, “Compared to both me-177 dia, there was more significant growth using MOD broth after 15 hours”. which statistic test was applied?
Line 189, what is the need of doing co-culture? that technique is generally used for microorganisms that are not harmful to humans such as probiotics, as it will arise a possibility of direct use of the strain as an antimicrobial agent rather than its metabolites.
Line 192, How do you measure the inhibition zone in broth?
Add a footnote to Table 3.
Line 234, it will be easy to run TLC or biochemical test to screen protein, lipids and sugars
Line 236, extraction yield, purification process? preliminary characterization, the fractionation process?
Fig 2, 3, the vertical axis should be “inhibition diameter” rather than “antimicrobial activity”
Fig 4, the vertical axis should be “microbial load (Log10 CFU/mL)” rather than “Log10 CFU/mL”
Fig 5, the vertical axis should be “inhibition diameter” rather than “antimicrobial activity”
Fig 6, provides IC50 for antioxidant activity
Fig 7, revise axis title. Also, explain why above 0.44 mg/mL P. aeruginosa load increased significantly.
Discussion
Line 266 Knowing this issue why have you decided to select such strains?
Line 269, do these authors use the same techniques as those reported in this study? you cannot state that observations are similar.
Authors should check yeast growth, might be E. coli has inhibited their growth
Line 293, explain why only 4 isolates amongst the 32?
Line 310, what means SLC or it is CFS? Also, it was observed that CB and CFS activity do not vary significantly, provide an explanation.
Line 310, why only a Gram-negative strain was used in this study for testing?
It is well known that the nature of antimicrobial metabolites as well as their antimicrobial mechanism vary with growth duration, further studies such as SEM, MBC etc; are needed because the observed activity at 24 h could be only an inhibition while that at 48 a bactericidal action.
Materials and methods
Provide the full information of reference strains. add the origin of those strains
It will be very important to mention the origin of Bacillus strains as it can impact the antimicrobial activity.
Also, provide the full name of the Bacillus strains, you previously mentioned that the strains come from a culture collection. Non-identified strains cannot be stored in a collection.
Line 420 Provide a brief description of the method
Why two different methods for the antagonistic activity of yeast and bacteria? as it is known that the method used can influence the results.
Line 426 stationary is not scientific, how it was assessed?
I suggest that authors avoid using the term optimization. What you did was to assess the effect of culture conditions and medium composition on microbial growth
Line 436-440, revise the sentence.
Line 440, also mention the inoculum load
Line 457, what is the carbon source in LB? and what is the carbon source in TSB? then, which carbon source was replaced?
It is not optimum, correct this term in the whole paper
Line 451 Bacillus are known for their heat resistance, why 45°C was not selected.? Are you aware that interesting antimicrobial metabolites are produced during sporulation?
The effect of either carbon or nitrogen source is generally assessed using a mineral salt medium where only one carbon source is added. If we considered the culture media used in this study, Luria Bertani broth (LB) (tryptone, 10 g/L; yeast extract, 5 g/L; NaCl, 10 437 g/L): Tryptone Soy Broth (TSB) (tryptone, 17 g/L; soybean, 3 g/L; glucose, 2.5 g/L; NaCl, 438 5.0 g/L; K2HPO4, 2.5 g/L) and Yeast Extract-Peptone-Glucose broth (YPG) (yeast extract, 439 5.0 g/L, peptone, 10 g/L; glucose 30 g/L), Can you specify the carbon source or nitrogen source that was replaced? considering the case of TSB, soybean (nitrogen and carbon source) will be replaced by what?
Line 470, what is the need to assess biomass?
Line 482-483, revise the sentence
Line 485, why latency phase culture for the test? provide clarification
Line 504, what was used as solvent? if it is water, what was the impact of the rotary evaporator on the activity of the concentrated CFS?
After going through the whole section 4.7. i did not find anything related to the optimization process, kindly provide more data and clarification
Line 497, mention these cultures here
Line 501, do the authors check the presence of residual sugar? as dialysis is not a suitable method to remove sugar from a mixture.
Line 504, what was used as solvent? if it is water, what was the impact of the rotary evaporator on the activity of the concentrated CFS?
Line 586, the screening and antimicrobial test as well as characterization of antimicrobial metabolites were performed prior to molecular identification, present the results accordingly.
Comments on the Quality of English LanguageHave your paper read by native English speakers.
Author Response
Dear Reviewer,
We greatly appreciate your time in reviewing this manuscript. Your comments and observations on the submitted work are valued, as we believe they are pertinent to enhancing the article. Please see below for detailed responses, and the corresponding revisions/corrections are highlighted in blue in the attachment.
The English grammar has been reviewed, and changes are indicated in blue.
Other comments:
Question 1:
Title
the term fraction is not appropriate, change it.
“isolated from Peru”, Peru is not a sample where microorganisms can be isolated; revise that, and mention the samples from which the strain was isolated.
Reply: Thank you very much for taking the time to review this manuscript. We appreciate your comments and observations. The title was modified according to suggestions: Antimicrobial activity of Bacillus amyloliquefaciens BS4 against Gram-negative pathogenic bacteria.
Question 2:
Abstract
Provide criteria for selecting the Gram-negative strain Escherichia coli ATCC 25922 for your research and no other pathogens.
Reply: Our research group is focused on identifying molecules with activity against Gram-negative bacteria, particularly those that are resistant to antimicrobials. Initially, we conducted biotechnological screening using E. coli ATCC 25922, and subsequently expanded our testing to include resistant clinical isolates
Question 3:
Line 14, separates the general objective of the work and the methodology
Reply: Thanks. It was separated for better understanding.
Question 4:
Line 22, these strains should be included in the methodology
Reply: Thanks for the observation. These strains are described in the methodology (Line 533-541).
Question 5:
Line 26-27,” the production is influenced by factors such as.. .” revise that sentence and add all the significant factors arising from the screening design performed in your study.
Reply: Thanks for the suggestion. We added in Line 26-27 that the production is influenced by carbon and nitrogen sources.
Question 6:
line 30, add the concentration of broth ingredients like starch, tryptone, and magnesium sulfate.
Reply: Thanks for the suggestion. We added in Line 27 the concentration of ingredients: starch (2.5g/L), tryptone (20g/L), and magnesium sulfate (0.2g/L).
Question 7:
Bacterial resistance cannot stand as a keyword, as resistance was not assessed in this study. remove. Add Bacillus sp. BS4
Reply: Thanks for the suggestion. We added as a keyword “Bacillus amyloliquefaciens BS4”.
Introduction
Question 8:
line 56, provide a paragraph that might justify why yeast strains should be used in this study.
Reply: In line 60-63, we add a paragraph about the importance of yeast.
Question 9:
Line 85, there are some interesting antimicrobial metabolites produced by microorganisms that remain attached to cell wall and need specific treatment to be detached. Why these metabolites were not considered in this study?
Reply: It is planned to carry out subcellular fractionation of different bacteria with antagonist activity (Research in preparation).
Results
Question 10:
Line 101, these tests might be insufficient to assign the isolate to Bacillus genus
Reply: We conducted both phenotypic and molecular identification of the isolates. The 16S rRNA gene was sequenced for phylogenetic analysis, and complete genome sequencing was performed as part of an ongoing manuscript.
Question 11:
Table 1, perform LSD to compare means. This will be the scientific base to select BS4 as the main important strain. also, mention E. coli in the table title.
Reply: was added.
Question 12:
Line 119, as there was no significant difference you cannot select TSB
Reply: Thanks for the suggestion. The line 119 was modified.
Question 13:
Table 2, add LSD superscript to compare means according to time and strains
Reply: was added
Question 14:
What happens above 72 h? it will bring more information on the growth phase and ease the selection of optimal conditions. Experiment should have been performed at 96 h.
Reply: According to Usta & Demirkan (2013), the antimicrobial activity of the Bacillus genus is typically observed within the timeframe of 24 to 72 hours. Additionally, reports on Bacillus growth kinetics commonly cover the period of 0 to 48 hours.
Question 15:
Line 177, “Compared to both media, there was more significant growth using MOD broth after 15 hours”. which statistic test was applied?
Reply: Thank you for your suggestions. We reviewed and corrected this paragraph.
Question 16:
Line 189, what is the need of doing co-culture? that technique is generally used for microorganisms that are not harmful to humans such as probiotics, as it will arise a possibility of direct use of the strain as an antimicrobial agent rather than its metabolites.
Reply: The co-culture is a standard methodology used as a strategy to induce the production of antimicrobial metabolites (Marmann et al. 2014). The interaction or competition encountered during co-cultivation is known to trigger a notable increase in the production of compounds that are constitutively present, and/or the accumulation of cryptic compounds that remain undetected in the isolated cultures of the producing strain.
Question 17:
Line 192, How do you measure the inhibition zone in broth?
Reply: Thank you for your suggestions. We have specified this in the methodology part.
Question 18:
Add a footnote to Table 3.
Reply: was added
Question 19:
Line 234, it will be easy to run TLC or biochemical test to screen protein, lipids and sugars.
Reply: Thank you for your observation. These essays have not yet been performed.
Question 20:
Line 236, extraction yield, purification process? preliminary characterization, the fractionation process?
Reply: The text has been revised for clarity. No purification process was undertaken.
Question 27:
Fig 2, 3, the vertical axis should be “inhibition diameter” rather than “antimicrobial activity”
Reply: We changed it.
Question 28:
Fig 4, the vertical axis should be “microbial load (Log10 CFU/mL)” rather than “Log10 CFU/mL”
Reply: Thanks for the suggestion. It was corrected.
Question 29:
Fig 5, the vertical axis should be “inhibition diameter” rather than “antimicrobial activity”
Reply: was corrected
Question 30:
Fig 6, provides IC50 for antioxidant activity
Reply: Thank you for your observation. These experiments have not been conducted yet. We only conducted a qualitative assessment of antioxidant activity.
Question 31:
Fig 7, revise axis title. Also, explain why above 0.44 mg/mL P. aeruginosa load increased significantly.
Reply: The assay was read for absorbance, probably the pigment that Pseudomonas produces was also read.
Discussion
Question 32:
Line 266 Knowing this issue why have you decided to select such strains?
Reply: These yeast strains were selected due to their isolation from environments characterized by extreme conditions and high levels of stress. Our objective was to investigate whether these strains produce bioactive metabolites.
Question 33:
Line 269, do these authors use the same techniques as those reported in this study? you cannot state that observations are similar. Authors should check yeast growth, might be E. coli has inhibited their growth
Reply: The comparison of line 371 was carried out with research that used the same technique that was used in the present study.
Question 34:
Line 293, explain why only 4 isolates amongst the 32?
Reply: Thanks for your comment. As a screening was conducted against E. coli ATCC 25922, only four out of the 32 strains exhibited antagonistic activity against this reference strain.
Question 35:
Line 310, what means SLC or it is CFS? Also, it was observed that CB and CFS activity do not vary significantly, provide an explanation.
Reply: Thanks for your comment. It was corrected. CFS is the sole abbreviation for cell-free supernatant.
Question 37:
Line 310, why only a Gram-negative strain was used in this study for testing?
Reply: Our research group is focused on identifying molecules with activity against Gram-negative bacteria, particularly those that are resistant to antimicrobials. Initially, we conducted biotechnological screening using E. coli ATCC 25922, and subsequently expanded our testing to include resistant clinical isolates
Question 38:
It is well known that the nature of antimicrobial metabolites as well as their antimicrobial mechanism vary with growth duration, further studies such as SEM, MBC etc; are needed because the observed activity at 24 h could be only an inhibition while that at 48 a bactericidal action.
Reply: Thank you for your observation. The test was carried out using the well diffusion method, and the inhibition zones were maintained for over 72 hours, thus demonstrating a bactericidal effect.
Materials and methods
Question 39:
Provide the full information of reference strains. add the origin of those strains
Reply: The reference strains belong to the strain collection of the Laboratory of Environmental Microbiology and Biotechnology - Faculty of Biological Science, Universidad Nacional Mayor de San Marcos (UNMSM).
Question 40:
It will be very important to mention the origin of Bacillus strains as it can impact the antimicrobial activity.
Reply: The strains originated from agricultural soil in Cerro de Pasco, Peru, and have been submitted to GenBank with the following IDs: PP396157 (BS3) and PP396158 (BS4).
Question 41:
Also, provide the full name of the Bacillus strains, you previously mentioned that the strains come from a culture collection. Non-identified strains cannot be stored in a collection.
Reply: The strains originate from collections from prior studies currently under investigation, and they belong to the laboratory's strain collection.
Question 42:
Line 420 Provide a brief description of the method
Reply: The method was described as more extended.
Question 43:
Why two different methods for the antagonistic activity of yeast and bacteria? as it is known that the method used can influence the results.
Reply: The assay was conducted according to the methodology described in the study by Acuña-Fontecilla et al. (2017).
Question 44:
Line 426 stationary is not scientific, how it was assessed?
Reply: Thank you. It was changed for “cultures from 18 to 24 hours”.
Question 45:
I suggest that authors avoid using the term optimization. What you did was to assess the effect of culture conditions and medium composition on microbial growth
Reply: We corrected it.
Question 47:
Line 436-440, revise the sentence.
Reply: We checked and corrected it.
Question 48:
Line 440, also mention the inoculum load
Reply: Thank you. We add it.
Question 49:
Line 457, what is the carbon source in LB? and what is the carbon source in TSB? then, which carbon source was replaced?
Reply: In the methodology, we detailed the composition of each culture medium. In this instance, TSB was identified as the best broth culture. While other components remained constant, the carbon source was modified.
Question 50:
It is not optimum, correct this term in the whole paper
Reply: Thank you. We checked and changed.
Question 51:
Line 451 Bacillus are known for their heat resistance, why 45°C was not selected.? Are you aware that interesting antimicrobial metabolites are produced during sporulation?
Reply: Thanks for your suggestions. This essay has not yet been performed.
Question 52:
The effect of either carbon or nitrogen source is generally assessed using a mineral salt medium where only one carbon source is added. If we considered the culture media used in this study, Luria Bertani broth (LB) (tryptone, 10 g/L; yeast extract, 5 g/L; NaCl, 10 437 g/L): Tryptone Soy Broth (TSB) (tryptone, 17 g/L; soybean, 3 g/L; glucose, 2.5 g/L; NaCl, 438 5.0 g/L; K2HPO4, 2.5 g/L) and Yeast Extract-Peptone-Glucose broth (YPG) (yeast extract, 439 5.0 g/L, peptone, 10 g/L; glucose 30 g/L), Can you specify the carbon source or nitrogen source that was replaced? considering the case of TSB, soybean (nitrogen and carbon source) will be replaced by what?
Reply: Thank you. The paragraph was rewritten to explain the one-factor-at-a-time method.
Question 53:
Line 470, what is the need to assess biomass?
Reply: Thanks for your observation. The aim was to assess microbial growth (biomass) by altering various components of the culture medium; nevertheless, comparable results were achieved.
Question 54:
Line 482-483, revise the sentence
Reply: We checked it.
Question 55:
Line 485, why latency phase culture for the test? provide clarification
Reply: The objective of the co-culture trial was to utilize Escherichia coli as an inducer of antagonistic metabolites; thus, its metabolic activity was not specifically required.
Question 56:
Line 504, what was used as solvent? if it is water, what was the impact of the rotary evaporator on the activity of the concentrated CFS?
Reply: Due to the presumed nature of the molecules (peptides) present in CFS, distilled water was chosen as a solvent. This choice aimed to minimize potential alterations to the activity and composition of the molecules. By using distilled water, any observed antagonistic activity could be confidently attributed to the molecules themselves rather than to the solvent.
Question 57:
After going through the whole section 4.7. i did not find anything related to the optimization process, kindly provide more data and clarification
Reply: Thanks. The term “optimization” was changed. in the text.
Question 58:
Line 497, mention these cultures here
Reply: Thanks for your suggestion. These cultures are mentioned in detail in the methodology (Line 533-541).
Question 59:
Line 501, do the authors check the presence of residual sugar? as dialysis is not a suitable method to remove sugar from a mixture.
Reply: Thank you for your clarification. This test was not carried out.
Question 60:
Line 586, the screening and antimicrobial test as well as characterization of antimicrobial metabolites were performed prior to molecular identification, present the results accordingly.
Reply: Thank you for your clarification. All tests described were conducted before molecular identification. Furthermore, strains exhibiting higher antagonist activity will undergo sequencing to further investigate their genome.
Reviewer 2 Report
Comments and Suggestions for Authors
The manuscript by Ana Palacios-Rodriguez and colleagues describes a study where they screened different microorganisms to obtain strains with antagonistic activity against Escherichia coli ATCC 25922. The study also evaluated the effect of culture conditions on improving the production of antimicrobial metabolites and determined the spectrum of action, antioxidant, and antihemolytic activity of the cell-free supernatant of the selected strain, BS4.
Comments
1. The study's results and discussion sections mention the significance of the research outcomes; however, they lack proper statistical analysis. For instance, Tables 2, 3, 4, and Figures 3 and 5 do not include statistical analysis. It is recommended that the statistical analysis results be added to tables and figures. Also, biased test results should not overclaim or overstate the study outcomes.
2. In Table 2, why did the YPG and LB media reduce the antimicrobial activity of BS4? In addition, I am wondering why the YPG and LB media were chosen to be compared with the TSB medium, which was already used to screen bacterial strains, in the first place. Please explain the reasons for this choice to reduce bias.
3. Figure 3 should be presented in three separate figures to be consistent with the one factor-at-a-time test.
Comments on the Quality of English LanguageThe language and structure of the manuscript are easy to understand. A few typos should be improved, such as "horas" in Line 97.
Author Response
Dear Reviewer,
We greatly appreciate your time in reviewing this manuscript. Your comments and observations on the submitted work are valued, as we believe they are pertinent to enhancing the article. Please see below for detailed responses, and the corresponding revisions/corrections are highlighted in blue in the attachment.
The English grammar has been reviewed, and changes are indicated in blue.
Question 1:
The study's results and discussion sections mention the significance of the research outcomes; however, they lack proper statistical analysis. For instance, Tables 2, 3, 4, and Figures 3 and 5 do not include statistical analysis. It is recommended that the statistical analysis results be added to tables and figures. Also, biased test results should not overclaim or overstate the study outcomes.
Reply: We correct it.
Question 2:
In Table 2, why did the YPG and LB media reduce the antimicrobial activity of BS4? In addition, I am wondering why the YPG and LB media were chosen to be compared with the TSB medium, which was already used to screen bacterial strains, in the first place. Please explain the reasons for this choice to reduce bias.
Reply: These media were used following the assay by Sa-Uth et al. (2018). Additionally, YPG broth is also employed for Bacillus in enzyme production and other metabolites.
Question 3:
Figure 3 should be presented in three separate figures to be consistent with the one factor-at-a-time test.
Reply: We separate figures.
Comments on the Quality of English Language
The language and structure of the manuscript are easy to understand. A few typos should be improved, such as "horas" in Line 97.
Reply: The entire document was reviewed and the changes are noted in blue
Reviewer 3 Report
Comments and Suggestions for Authors
“Antimicrobial activity of secreted fractions of microorganisms isolated from Peru”
By Palacios-Rodriguez et al.
The paper has some significant and important results presented. However, the manuscript appears to have been written in haste. Experiments have been properly designed but poorly presented with a lot of information missing and some contradictory statements. The Discussion section is a little long but all information in it is relevant. Some more comments on the paper are mentioned below.
Line 63: “biosynthetic genes (BCGs)” Why use an uncommon abbreviation that is used only twice in the whole manuscript?
Line 64: “resulting in helpful finding new metabolites” Grammatically incorrect. Not sure what the authors intended to say. Possible suggestion: “resulting in discovery of new metabolites”
Line 67: “These silenced genes are called……pathways” One gene cannot make a pathway.
Line 91: Change “Antagonistic Bacterial” to “Antagonistic Bacteria”
Line 95: “bacterial culture (CB)” Same comment as for Line 425.
Line 97: Change “horas” to “hours”
Line 103 and line 311: “SLC” Full form of all abbreviations should be provided at the first mention. This is written later in line 444 as cell-free supernatant. However, there are two problems. The full form, “cell-free supernatant” has also been used numerous times in the manuscript. Secondly, a different abbreviation, CFS has been mentioned for the same term (line 96 and 425, 521)
Line 101: “short” Meaning of this is not clear. If please specify a size scientifically.
Line 107: Magnification for Figure 1a is not mentioned.
Line 109: “scale bar” There is no scale bar shown.
Line 113: “antimicrobial activity” It is not clear and not mentioned whether CB or CFS was used for this experiment. From line 116 it appears that CFS was used, but this should be clearly mentioned. Same comment about line 121 (Table 2).
Line 113-115: “The maximum amount of biomass ….. after 24 hours in TSB broth” I see the biomass to be almost similar in all of them. So instead of focusing on this minor difference, it will be better to state that for similar biomasses, TSB medium gave the highest antimicrobial activity. This can be seen even better if zone of inhibition per CFU (or Log10CFU) is presented.
Line 125: “pH 5 to pH 9” Are there two variables here, pH and growth? Was growth (OD600) or biomass measured? Does antimicrobial activity correlate with growth? What are being optimized here are conditions of growth and not that of antibiotic production. If so, this result is not surprising. Same comments about temperature and agitation. For temperature, biomass at 28 and 30 oC are mentioned but not the others. For agitation, it is not clear if biomass was measured for all or only 150 and 180 rpm. It is not clear why the activity at 37 oC is lower. Was there less growth? If not, this will probably be the only data point for which activity is lower for the same biomass. Is there an explanation?
Line 141: “Effect of Carbon, Nitrogen, and inorganic salts” Please mention what were the sources of the other two when a third one was varied.
Line 148: “activity decreased when both sources were used”. Was biomass measured? Did it also decrease or remain the same?
Line 151: “various nitrogen sources were tested”. Was biomass measured? Did it vary or remain the same?
Line 162: “modified medium (MOD) ingredients” Besides the three mentioned, what other components were present in MOD? In line 170 it is mentioned “TSB medium such as basal medium” Not sure what this means. Is it dextrose-free TSB? Using this medium how can the nitrogen source be changed?
Line 197: “with thermally inactive …..cells” How many inactive cells were added? Equal to what is added to the second tube at the start of growth, or equal to the final E. coli cell concentration achieved in the second tube at the end of the growth?
Line 201: “sown” Do you mean, “spread”? I have never heard the term sowing bacteria on a plate.
Line 201: “was bactericidal because it was sown on a plate with MacConkey agar, and the presence of colonies was not observed on the plate (data not shown).” If you discuss a result, do that scientifically and with proper details. Otherwise, there is no need to mention it especially if the data is not shown. It is not clear how MacConkey agar is related to bactericidal activity. What was spread on the plate? After what experiment?
Line 218: “TSB modified” Is this the same as MOD? Why use two names for the same thing?
Line 220: Section 2.6.2. This is one of the most poorly written sections. Please make it better.
Line 221: “The results indicated” Please explain how the results indicate.
Line 222: “antihemolytic and antioxidant activity”. Why are these two activities always discussed together?
Line 222: “the supernatant” Which supernatant? Is FC the supernatant? If so, why use multiple names for the same thing? Why is the activity of FC different from the 3KDa fraction?
Line 227: “3Kda fraction”. What is meant by 3KDa fraction? Is it > 3 KDa or <3 KDa? Whichever it is, where is the data for the other one? (after filtration and before filtration)
Line 227: “Trolox” This is the first and only mention of the word. Please write what it is and why it is used here.
Line 225, Figure 6: What are the three sample in each row? What is there is the fifth row? Why is there only one sample in the fifth row? The handwriting for this (ABTS) is hardly legible. Why is the figure labeled with handwritten symbols? Please make the figure presentable.
Line 225, Figure 6 Panel (b): What is one supposed to see in this figure? The handwriting in this panel is hardly visible and not legible. Please explain what the different rows and columns represent.
Line 230: “Treatment” How long was the treatment?
Line 233: “antibacterial compounds derived from protein compounds” to “antibacterial compounds derived from protein”
Line 237: “lyophilized”. Materials and Methods says that it was rotary evaporated. Please don’t write contradictory statements especially for experimental procedure.
Line 241 Figure 7: Y- axis says 595 nm. In Line 479 it is mentioned as 600 nm.
Line 241: Figure 7: At what time of growth were these OD measurements taken?
Line 242: “Growth curve” This is not called a growth curve, which is usually a plot of growth Vs time.
Line 425, line 95 and other places: “bacterial culture (CB)” This term is misleading. Bacterial culture usually mean cells plus the medium. In other word, does CB include CFC? Please explain the procedure properly. It is my guess that the bacterial culture was centrifuged. The supernatant was filtered and used as CFC while the pellet was resuspended in fresh medium or buffer (please specify) and used as CB. Please confirm if this is correct. If this has only cells then the appropriate term to use will be “bacterial cells (CB)” instead of “bacterial culture (CB)” here and other places. If CB contains (cells + CFS), then CB should always have more activity than CFS. However, results in Table 2 do not support that (see 72 h results)
Line 461: “selected base medium” Please specify what this base medium is. I guess, it does not contain any C, N or salt. What does it contain? When C source is varied, what is the N and salt source? Similarly, when one of the other two is varied, what provides the remaining two?
Line 473: “10% (v/v) aliquot” Aliquots are measured by mass or volume and not as percentages. It is not clear, percent of what? The source culture, or the final medium? I guess, you transferred 5 ml aliquot to 45 ml medium.
Line 478: “measured by spectrophotometry at 600 nm” A600 values are not reported anywhere in the manuscript. Biomass and growth shown in Figure 4 and Table 2 are in CFU/ml. Is the CFU calculated from A600 or is it obtained by spreading serial dilutions on plates? If it is calculated from A600, then it is not justified to report it as CFU, you can simply report as A600. If done by spreading serial dilutions and counting colonies, that method should be mentioned.
Line 501: “concentrated against distilled water using a dialysis bag” This is not a concentration step. Maybe “concentrated” can be changed to “dialyzed”
Line 503: Change “by dialysis bag” to “from dialysis bag”
Line 504: “the eluate was again concentrated with a rotary evaporator” Aqueous solutions are usually not concentrated by rotary evaporation. How long did it take? Was the temperature increased? What were the initial or final volumes, or, how many fold concentration was obtained? Why was it not freeze dried (lyophilized)? Here you mention that it was rotary evaporated but in the =Results Section (Line 237) you mention that it was lyophilized.
Line 508: “GR” Full form of this abbreviation has not been mentioned anywhere.
Line 509: “the supernatant was read” Was there a precipitate? How was the supernatant separated from the precipitate? By centrifugation or by gravity?
Line 512: “For this assay” Which assay? Why are two assays hemolytic and antioxidant activities discussed together? The results are shown in two separate panels in Figure 6.
Line 513 and 403: “ABTS” Full form of this abbreviation has not been mentioned anywhere.
Line 516: Change “were investigated” to “on the CFS were investigated”
Line 518-519: “The cell-free……respectively.” This sentence does not have a verb.
Line 519: Change “buffer as” to “buffer were used as”
Line 518: “The cell-free supernatant without treatment” Not clear what this means. The CFS is not treated, cells are.
Line 522, 525, 531, 533: “SLC” What is SLC? Same as CFS? Why use two names?
Line 545: “595 nm” In line 479 this is written as 600 nm.
Line 546: “positive death control was the antibiotic chloramphenicol”. Never heard of “positive death control”. Why not “positive control”? By the way, chloramphenicol is bacteroiostatic and not bactericidal and cannot be a positive control for death but is fine for use in MIC experiments.
Line 549, 554: “reactivated” Not clear what is being reactivated. Do you mean “inoculated”?
Line 554-557: “Bacterial ……… (Promega)” This whole paragraph is copied from the previous section for no reason and serves no purpose here.
Line 577, 579, 581: “will be” Why future tense?
Line 600: Change “yeasts as” to “yeasts such as”
Line 602: Change “proteinaceous was” to “proteinaceous nature was”
Throughout the manuscript, for “micro” use m in symbol font instead of u
Comments on the Quality of English Language
Some minor corrections needed as mentioned above.
Author Response
Dear Reviewer,
We greatly appreciate your time in reviewing this manuscript. Your comments and observations on the submitted work are valued, as we believe they are pertinent to enhancing the article. Please see below for detailed responses, and the corresponding revisions/corrections are highlighted in blue in the attachment.
The English grammar has been reviewed, and changes are indicated in blue.
Question 1:
Line 63: “biosynthetic genes (BCGs)” Why use an uncommon abbreviation that is used only twice in the whole manuscript?
Reply: Thanks for your comment. It was corrected.
Question 2:
Line 64: “resulting in helpful finding new metabolites” Grammatically incorrect. Not sure what the authors intended to say. Possible suggestion: “resulting in discovery of new metabolites”
Reply: Thanks for your comment. It was corrected.
Question 3:
Line 67: “These silenced genes are called……pathways” One gene cannot make a pathway.
Reply: Thank you for your observation. It was corrected in lines 76-78.
Question 4:
Line 91: Change “Antagonistic Bacterial” to “Antagonistic Bacteria”
Reply: Thank you. It was checked and corrected.
Question 5:
Line 95: “bacterial culture (CB)” Same comment as for Line 425.
Reply: Thank you. It was checked and corrected.
Question 6:
Line 97: Change “horas” to “hours”
Reply: Thank you. It was checked and corrected.
Question 7:
Line 103 and line 311: “SLC” Full form of all abbreviations should be provided at the first mention. This is written later in line 444 as cell-free supernatant. However, there are two problems. The full form, “cell-free supernatant” has also been used numerous times in the manuscript. Secondly, a different abbreviation, CFS has been mentioned for the same term (line 96 and 425, 521)
Reply: We changed, it was a mistake of translation.
Question 8:
Line 101: “short” Meaning of this is not clear. If please specify a size scientifically.
Reply: Thanks for your comment. The text was modified.
Question 9:
Line 107: Magnification for Figure 1a is not mentioned.
Reply: We checked and corrected it.
Question 10:
Line 109: “scale bar” There is no scale bar shown.
Reply: Thank you for your comment. It was corrected.
Question 11:
Line 113: “antimicrobial activity” It is not clear and not mentioned whether CB or CFS was used for this experiment. From line 116 it appears that CFS was used, but this should be clearly mentioned. Same comment about line 121 (Table 2).
Reply: Thank you for the observation. We used CFS, which is specified more extended in the methodology (Line 575-578).
Question 12:
Line 113-115: “The maximum amount of biomass ….. after 24 hours in TSB broth” I see the biomass to be almost similar in all of them. So instead of focusing on this minor difference, it will be better to state that for similar biomasses, TSB medium gave the highest antimicrobial activity. This can be seen even better if zone of inhibition per CFU (or Log10CFU) is presented.
Reply: Thank you for your correction. We checked and corrected it.
Question 13:
Line 125: “pH 5 to pH 9” Are there two variables here, pH and growth? Was growth (OD600) or biomass measured? Does antimicrobial activity correlate with growth? What are being optimized here are conditions of growth and not that of antibiotic production. If so, this result is not surprising. Same comments about temperature and agitation. For temperature, biomass at 28 and 30 oC are mentioned but not the others. For agitation, it is not clear if biomass was measured for all or only 150 and 180 rpm. It is not clear why the activity at 37 oC is lower. Was there less growth? If not, this will probably be the only data point for which activity is lower for the same biomass. Is there an explanation?
Reply: The experiments assessing the effects of temperature, pH, and agitation were assessed to identify the best conditions for maximal antagonist activity. Biomass measurements were obtained for all conditions (data not presented). Regarding temperature, the highest level of antagonist activity was observed between 28 and 30°C, whereas at 37°C, both the activity and biomass decreased.
Question 14:
Line 141: “Effect of Carbon, Nitrogen, and inorganic salts” Please mention what were the sources of the other two when a third one was varied.
Reply: When this experiment was conducted following the methodology outlined by Sa-Uth et al. (2018), the one-factor-at-a-time approach was utilized. For example, while varying the carbon source, the nitrogen sources (soy peptone and tryptone), and the mineral salts (NaCl and K2HPO4) were kept constant.
Question 15:
Line 148: “activity decreased when both sources were used”. Was biomass measured? Did it also decrease or remain the same?
Reply: Biomass was measured in CFU/mL. There was no significant variation in biomass
Question 16:
Line 151: “various nitrogen sources were tested”. Was biomass measured? Did it vary or remain the same?
Reply: Biomass was measured in CFU/mL
Question 17:
Line 162: “modified medium (MOD) ingredients” Besides the three mentioned, what other components were present in MOD? In line 170 it is mentioned “TSB medium such as basal medium” Not sure what this means. Is it dextrose-free TSB? Using this medium how can the nitrogen source be changed?
Reply: Thank you. The components of MOD broth were starch, tryptone, and magnesium sulfate. The TSB broth was the selected broth based on which the variation of carbon, nitrogen, and mineral salt sources was made.
Question 18:
Line 197: “with thermally inactive …..cells” How many inactive cells were added? Equal to what is added to the second tube at the start of growth, or equal to the final E. coli cell concentration achieved in the second tube at the end of the growth?
Reply: A concentration of 0.5 OD of thermally inactivated E.coli cells was added according to the McFarland scale (1.5 x 108 cells/mL).
Question 19:
Line 201: “sown” Do you mean, “spread”? I have never heard the term sowing bacteria on a plate.
Reply: The paragraph was eliminated.
Question 20:
Line 201: “was bactericidal because it was sown on a plate with MacConkey agar, and the presence of colonies was not observed on the plate (data not shown).” If you discuss a result, do that scientifically and with proper details. Otherwise, there is no need to mention it especially if the data is not shown. It is not clear how MacConkey agar is related to bactericidal activity. What was spread on the plate? After what experiment?
Reply: The paragraph was eliminated.
Question 21:
Line 218: “TSB modified” Is this the same as MOD? Why use two names for the same thing?
Reply: Thank you for your comment. The use of the term MOD is now standardized for the medium that was modified during the tests
Question 22:
Line 220: Section 2.6.2. This is one of the most poorly written sections. Please make it better.
Reply: We checked and improved this paragraph.
Question 23:
Line 221: “The results indicated” Please explain how the results indicate.
Reply: Thank you. The paragraph was modified for better understanding.
Question 24:
Line 222: “antihemolytic and antioxidant activity”. Why are these two activities always discussed together?
Reply: Our research group is primarily focused on identifying antimicrobial molecules. However, in this study, we also evaluated additional properties to identify molecules that are not cytotoxic (non-hemolytic) and possess antioxidant activity.
Question 25:
Line 222: “the supernatant” Which supernatant? Is FC the supernatant? If so, why use multiple names for the same thing? Why is the activity of FC different from the 3KDa fraction?
Reply: FC means a complete fraction of the cell-free supernatant.
The activity of the complete fraction (FC) differs from that of the 3KDa fraction because the latter contains molecules with a molecular weight cut-off of 3KDa, while the FC contains molecules of all sizes. Currently, we are in the process of identifying the components of these fractions (work in progress).
Question 26:
Line 227: “3Kda fraction”. What is meant by 3KDa fraction? Is it > 3 KDa or <3 KDa? Whichever it is, where is the data for the other one? (after filtration and before filtration).
Reply: The 3KDa fraction is all molecules whose molecular mass is less than 3KDa. The data for the largest molecules are found in the complete fraction (CF).
Question 27:
Line 227: “Trolox” This is the first and only mention of the word. Please write what it is and why it is used here.
Reply: TROLOX is an antioxidant compound that is commonly used as a reference in antioxidant assays. In this case, TROLOX was the positive control in the antioxidant activity essay.
Question 28:
Line 225, Figure 6: What are the three sample in each row? What is there is the fifth row? Why is there only one sample in the fifth row? The handwriting for this (ABTS) is hardly legible. Why is the figure labeled with handwritten symbols? Please make the figure presentable.
Reply: Figure 6 was submitted with handwritten symbols to demonstrate its originality as requested by the magazine; However, we appreciate your suggestion and we could make the corresponding labels.
Question 29:
Line 225, Figure 6 Panel (b): What is one supposed to see in this figure? The handwriting in this panel is hardly visible and not legible. Please explain what the different rows and columns represent.
Reply: Thank you for your feedback. The legend of the figure describes the layout of the handwriting. The columns represent the replicates, while the rows correspond to the reagents and samples.
Question 30:
Line 230: “Treatment” How long was the treatment?
Reply: In the methodology, Line 684-687 “Enzyme effect” we mentioned the duration of the treatment
Question 31:
Line 233: “antibacterial compounds derived from protein compounds” to “antibacterial compounds derived from protein”
Reply: Thanks for your suggestion. It was corrected.
Question 32:
Line 237: “lyophilized”. Materials and Methods says that it was rotary evaporated. Please don’t write contradictory statements especially for experimental procedure.
Reply: For the MIC essay, we lyophilize the CFS to concentrate the sample, weigh it, and thus have an exact concentration to carry out the serial dilution.
Question 33:
Line 241 Figure 7: Y- axis says 595 nm. In Line 479 it is mentioned as 600 nm.
Reply: Thank you. It was corrected to 600nm in all the cases.
Question 34:
Line 241: Figure 7: At what time of growth were these OD measurements taken?.
Reply: Thank you for your comment. The description of the figure was changed for better understanding. The readings were taken 24 hours after confronting the bacteria against CFS.
Question 35:
Line 242: “Growth curve” This is not called a growth curve, which is usually a plot of growth Vs time.
Reply: Thank you for your comment. The description of the figure was changed for better understanding.
Question 36:
Line 425, line 95 and other places: “bacterial culture (CB)” This term is misleading. Bacterial culture usually mean cells plus the medium. In other word, does CB include CFC? Please explain the procedure properly. It is my guess that the bacterial culture was centrifuged. The supernatant was filtered and used as CFC while the pellet was resuspended in fresh medium or buffer (please specify) and used as CB. Please confirm if this is correct. If this has only cells then the appropriate term to use will be “bacterial cells (CB)” instead of “bacterial culture (CB)” here and other places. If CB contains (cells + CFS), then CB should always have more activity than CFS. However, results in Table 2 do not support that (see 72 h results).
Reply: The term "Bacterial culture (CB)" refers to the combination of cells (biomass) and cell-free supernatant (CFS). However, in Table 2, the assay was conducted solely with the cell-free supernatant (CFS).
Question 37:
Line 461: “selected base medium” Please specify what this base medium is. I guess, it does not contain any C, N or salt. What does it contain? When C source is varied, what is the N and salt source? Similarly, when one of the other two is varied, what provides the remaining two?
Reply: As an initial step, the selection of the culture medium was conducted, with TSB determined as the best broth. Subsequently, the impact of carbon, nitrogen, and mineral salt sources was assessed. To manipulate these components, the TSB medium was prepared using individual constituents. Each component (carbon, nitrogen, and salts) was varied independently while keeping the other two constants.
Question 38:
Line 473: “10% (v/v) aliquot” Aliquots are measured by mass or volume and not as percentages. It is not clear, percent of what? The source culture, or the final medium? I guess, you transferred 5 ml aliquot to 45 ml medium.
Reply: We changed 10%(v/v) to an inoculum of 5 mL of this culture.
Question 39:
Line 478: “measured by spectrophotometry at 600 nm” A600 values are not reported anywhere in the manuscript. Biomass and growth shown in Figure 4 and Table 2 are in CFU/ml. Is the CFU calculated from A600 or is it obtained by spreading serial dilutions on plates? If it is calculated from A600, then it is not justified to report it as CFU, you can simply report as A600. If done by spreading serial dilutions and counting colonies, that method should be mentioned.
Reply: Plate counting was performed to determine CFU/mL. The corresponding methodology will be added.
Question 40:
Line 501: “concentrated against distilled water using a dialysis bag” This is not a concentration step. Maybe “concentrated” can be changed to “dialyzed”
Reply: It was corrected.
Question 41:
Line 503: Change “by dialysis bag” to “from dialysis bag”
Reply : It was corrected.
Question 42:
Line 504: “the eluate was again concentrated with a rotary evaporator” Aqueous solutions are usually not concentrated by rotary evaporation. How long did it take? Was the temperature increased? What were the initial or final volumes, or, how many fold concentration was obtained? Why was it not freeze dried (lyophilized)? Here you mention that it was rotary evaporated but in the =Results Section (Line 237) you mention that it was lyophilized.
Reply: The rotary evaporator was used to concentrate the eluate since the lyophilizer was under maintenance. The volume that was concentrated was 500uL and took approximately 3 hours.
Question 43:
Line 508: “GR” Full form of this abbreviation has not been mentioned anywhere.
Reply: Thanks for your comment. The term “GR” was changed to red blood cells.
Question 44:
Line 509: “the supernatant was read” Was there a precipitate? How was the supernatant separated from the precipitate? By centrifugation or by gravity?
Reply: The precipitation occurred naturally due to gravity when the Eppendorf tubes containing the samples were incubated at 37°C for 3 hours. Subsequently, the tubes were centrifuged, and the supernatant was collected for analysis. The absorbance of the supernatant was then measured at 405 nm using a spectrophotometer.
Question 45:
Line 512: “For this assay” Which assay? Why are two assays hemolytic and antioxidant activities discussed together? The results are shown in two separate panels in Figure 6.
Reply: Thank you for your comment. The term “for this assay” was changed to the name of the assay, in this case, antioxidant activity. Both tests are found in the same item because they were complementary tests to find molecules that are not cytotoxic (non-hemolytic) and that have antioxidant activity.
Question 46:
Line 513 and 403: “ABTS” Full form of this abbreviation has not been mentioned anywhere.
Reply: Thank you for your comment. We completed with the complete name.
Question 47:
Line 516: Change “were investigated” to “on the CFS were investigated”
Reply: It was corrected.
Question 48:
Line 518-519: “The cell-free……respectively.” This sentence does not have a verb.
Reply: Thanks for your suggestion. The paragraph was changed.
Question 49:
Line 519: Change “buffer as” to “buffer were used as”
Reply: Thanks for your suggestion. The paragraph was changed.
Question 50:
Line 518: “The cell-free supernatant without treatment” Not clear what this means. The CFS is not treated, cells are.
Reply: Thank you. CFS without treatment indicates that the supernatant was not subjected to enzymatic reactions, extreme temperatures, or surfactants during the experimental process.
Question 51:
Line 522, 525, 531, 533: “SLC” What is SLC? Same as CFS? Why use two names?
Reply: Apologies for these mistakes. It was changed at CFS which means cell-free supernatants
Question 52:
Line 545: “595 nm” In line 479 this is written as 600 nm.
Reply: Thanks for your comment. It was corrected to 600 nm.
Question 53:
Line 546: “positive death control was the antibiotic chloramphenicol”. Never heard of “positive death control”. Why not “positive control”? By the way, chloramphenicol is bacteriostatic and not bactericidal and cannot be a positive control for death but is fine for use in MIC experiments.
Reply: Thanks for your recommendation. Positive death control was changed to positive control.
Question 54:
Line 549, 554: “reactivated” Not clear what is being reactivated. Do you mean “inoculated”?
Reply: We changed “were reactivated” to “were cultured” in line 717.
Question 55:
Line 554-557: “Bacterial ……… (Promega)” This whole paragraph is copied from the previous section for no reason and serves no purpose here.
Reply: Thanks for your suggestion. This paragraph was eliminated.
Question 56:
Line 577, 579, 581: “will be” Why future tense?
Reply: These were corrected to past tense.
Question 57:
Line 600: Change “yeasts as” to “yeasts such as”
Reply: It was corrected.
Question 58:
Line 602: Change “proteinaceous was” to “proteinaceous nature was”
Reply: It was corrected.
Question 59:
Throughout the manuscript, for “micro” use m in symbol font instead of u
Reply: It was corrected.
Round 2
Reviewer 1 Report
Comments and Suggestions for Authors
The authors have incorporated most of the comments, although the answers to some are not satisfying.
Comments on the Quality of English LanguageThe paper must be revised by a native English speaker.